# GLEMOS: Benchmark for Instantaneous Graph Learning Model Selection

**Namyong Park**[1]*, **Ryan Rossi**[2], **Xing Wang**[1], **Antoine Simoulin**[1],
**Nesreen Ahmed**[3], **Christos Faloutsos**[4]
[1]Meta AI [2]Adobe Research [3]Intel Labs [4]Carnegie Mellon University

## Abstract

The choice of a graph learning (GL) model (*i.e.*, a GL algorithm and its hyperparameter settings) has a significant impact on the performance of downstream tasks. However, selecting the right GL model becomes increasingly difficult and time consuming as more and more GL models are developed. Accordingly, it is of great significance and practical value to equip users of GL with the ability to perform a *near-instantaneous* selection of an effective GL model without manual intervention. Despite the recent attempts to tackle this important problem, there has been no comprehensive benchmark environment to evaluate the performance of GL model selection methods. To bridge this gap, we present GLEMOS in this work, a comprehensive benchmark for instantaneous GL model selection that makes the following contributions. (i) GLEMOS provides extensive benchmark data for fundamental GL tasks, *i.e.*, link prediction and node classification, including the performances of 366 models on 457 graphs on these tasks. (ii) GLEMOS designs multiple evaluation settings, and assesses how effectively representative model selection techniques perform in these different settings. (iii) GLEMOS is designed to be easily extended with new models, new graphs, and new performance records. (iv) Based on the experimental results, we discuss the limitations of existing approaches and highlight future research directions. To promote research on this significant problem, we make the benchmark data and code publicly available at `https://namyongpark.github.io/glemos`.

## 1 Introduction

Graph learning (GL) methods [43, 48] have achieved great success across multiple domains and applications that involve graph-structured data [4, 10, 17, 21, 28, 29, 31, 34]. At the same time, previous studies [30, 40, 46] have shown that there is no universally good GL model that performs best across

*all* graphs and graph learning tasks. Therefore, to effectively employ GL given a wide array of available models, it is important to select the right GL model (*i.e.*, a GL algorithm and its hyperparameter settings) that will perform well for the given graph data and GL task.

Ideally, we would want to be able to select the best GL model for the given graph *near-instantaneously*, that is, without having to train or evaluate different models multiple times on the new graph, since even a few such training and evaluations might take a considerable

Figure 1: Via instantaneous graph learning model selection, the best model can be found without performing computationally expensive model training and evaluations.

*Correspondence: namyongp@meta.com

amount of time and resources (Figure 1). Enabling an instantaneous model selection for a completely new graph involves addressing several technical challenges, which includes modeling how well different GL methods perform on various graphs, and establishing a connection between the new graph and observed graphs, such that the best model for the new graph can be estimated in light of observed model performances on similar graphs.

**Problem Formulation.** With these considerations, we formally define this important problem, which we call *Instantaneous Graph Learning Model Selection*, and the related terms as follows.

*Graph:* Let $G = (V, E, X, Y)$ be a graph where $V \subseteq \mathbb{N}$ and $E = \{(i, j) \mid i, j \in V\}$ denote the sets of nodes and edges, respectively; $X$ denotes the input features, which can be node features ($X \in \mathbb{R}^{|V| \times F_N}$), edge features ($X \in \mathbb{R}^{|E| \times F_E}$), or a set of both, where $F_N$ and $F_E$ denote the dimension of corresponding input features; and $Y$ denotes node labels ($Y \in \mathbb{N}^{|V|}$) or edge labels ($Y \in \mathbb{N}^{|E|}$). Note that input features $X$ and labels $Y$ are considered optional since not all graphs have this information.

*Model:* A model $M$ refers to a GL method for the given GL task, such as link prediction, with specific hyperparameter settings. In general, a GL model consists of two components, namely, (graph embedding method, hyperparameters) and (predictor, hyperparameters), where the former produces a vector representation of the graph (*e.g.*, node embeddings) and the latter makes task-specific predictions (*e.g.*, link prediction) given the embeddings. The set $\mathcal{M}$ of models, from which the model selection is made, is normally heterogeneous, where the configuration of each model is unique in the choice of its two components and their hyperparameter settings.

*Performance Matrix:* Let $\mathbf{P} \in \mathbb{R}^{n \times m}$ be a matrix containing observed model performances, where $P_{ij}$ is the performance (*e.g.*, accuracy) of model $j$ on graph $i$. $\mathbf{P}$ can be sparse with missing entries.

**Problem 1** (Instantaneous Graph Learning Model Selection).

**Given**
- a training meta-corpus of $n$ graphs $\mathcal{G} = \{G_i\}_{i=1}^{n}$ and $m$ models $\mathcal{M} = \{M_j\}_{j=1}^{m}$ for a GL task (*e.g.*, link prediction and node classification):
  (1) performance matrices $\{\mathbf{P}_k\}_{k=1}^{\ell}$, *i.e.*, $\ell$ records of $m$ models' performance on $n$ graphs
  (2) input features of the graphs in $\mathcal{G}$ (if available)
  (3) configurations (*i.e.*, a GL method and its hyperparameter settings) of $m$ models in $\mathcal{M}$
- an unseen test graph $G_{\text{test}} \notin \mathcal{G}$

**Select**
- the best model $M^* \in \mathcal{M}$ for $G_{\text{test}}$ ***without*** training or evaluating any model in $\mathcal{M}$ on $G_{\text{test}}$.

**Status Quo and Our Contributions.** In recent years, several methods have been developed for an efficient selection of GL models. However, most of them cannot tackle Prob. 1 as they require multiple rounds of model training and evaluations; we review these methods in Sec. 2. Most recently, a subset of Prob. 1 was studied by MetaGL [30], which proposed a GL model selection technique that assumes plain graphs without input features, and operates without utilizing model configurations. A few recent works [5, 32, 46] also provide performances of graph neural networks (GNNs), although they cannot address Prob. 1. While the datasets used in [5, 30, 32, 46] are available, they fall short of being a comprehensive benchmark environment to study this significant problem due to the following reasons.

- **Limited GL Task and Data.** Focusing on link prediction, MetaGL [30] only provides link prediction performances, and does not support other widely-used tasks, such as node classification, which limits follow-up studies and use of the benchmark for different GL tasks. Also, other related works [5, 32, 46] are limited in terms of the number and diversity of graphs they cover (Table 1).
- **Limited Evaluation Settings.** Some important evaluation settings were not considered in MetaGL's benchmark, such as out-of-domain and small-to-large settings as we later describe, which can be useful in evaluating the performance of model selection techniques in different practical settings.
- **Limited Extensibility.** The sets of models and graphs are assumed to be fixed, and it is not easy to extend the benchmark with new graphs and models in a consistent and reproducible manner.

In this work, we address these limitations by developing a comprehensive benchmark for instantaneous graph learning model selection. Overall, the contributions of this work are as follows.

- **Extensive Benchmark Data with Multiple GL Tasks.** We construct a benchmark dataset that includes the performances of 366 models on 457 graphs over fundamental GL tasks, *i.e.*, link pre-

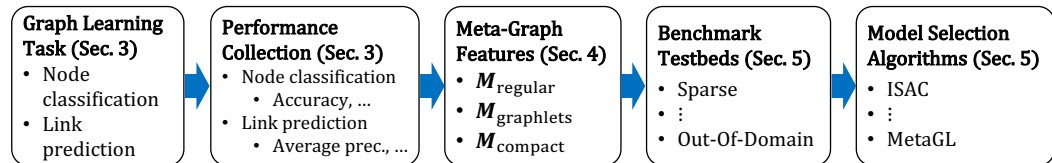

Figure 2: GLEMOS provides a comprehensive benchmark environment, covering the steps required to achieve effective instantaneous GL model selection, with multiple options for major building blocks.

diction and node classification, which is by far the largest benchmark for Prob. 1 to our knowledge. The benchmark also provides meta-graph features to capture the structural characteristics of graphs.

- **Comprehensive Evaluation Testbeds.** We evaluate ten representative methods for Problem 1, including both classical methods and deep learning-based ones, using multiple evaluation settings designed to assess the quality of model selection techniques from practical perspectives.
- **Extensible Open Source Benchmark Environment.** Our benchmark is designed to be easily extended with new models, new graphs, and new performance records. To promote further research on this significant problem, we make the benchmark environment publicly available.
- **Future Research Directions.** We discuss the limitations of existing model selection methods, and highlight future research directions towards an instantaneous selection of graph learning models.

After reviewing related work in Section 2, we present the proposed benchmark data and testbeds in Sections 3 to 5. Then we provide experimental results in Section 6, and conclude in Section 7.

## 2 Related Work

### 2.1 Model Selection

Model selection refers to the process of selecting a learning algorithm and its hyperparameter settings. In this section, we review existing model selection approaches, which we divide into two groups depending on whether they require model evaluations (*i.e.*, performance queries for the new dataset).

**Evaluation-Based Model Selection:** Most existing approaches to select machine learning models belong to this group, ranging from simple solutions, such as random search [2] and grid search [23], to more advanced and efficient ones that employ techniques such as adaptive resource allocation [22], early stopping [12], and Bayesian optimization [9, 33, 42]. Inspired by these advancements, several model selection methods were recently developed for graph learning (GL) models. To tackle challenges involved with GL model selection, these methods adapt existing ideas to GL models, such as reinforcement learning [11, 20, 50], evolutionary algorithm [3], Bayesian optimization [36], and hypernets [52], as well as developing techniques specific to graph data, *e.g.*, subgraph sampling [36] and graph coarsening [14]. Note that all of the above approaches cannot tackle the instantaneous GL model selection problem (Problem 1) as they rely on multiple model evaluations for performance queries of different combinations of GL methods and hyperparameter settings on the new dataset.

**Instantaneous Model Selection:** To select the best model without querying model performances on the new dataset, methods in this category typically utilize prior model performances or characteristic features of a dataset (*i.e.*, meta-features). A simple approach [1] finds the globally best model (*i.e.*, the one with the overall best performance over all observed datasets), and thus its model selection is independent of query datasets. This can be refined by narrowing the search scope to similar datasets, where dataset similarities are modeled in the meta-feature space, *e.g.*, using $k$-nearest neighbors [27] or clustering [18]. Another line of methods [30, 45, 49] take a different approach, which aims to predict the model performance on the given dataset by learning a function that maps meta-features into estimated model performances. Due to their ability to learn such a function in a data-driven manner, this second group of methods generally outperformed the first group in previous studies [30, 49]. While the above methods are one of the first efforts to achieve instantaneous GL model selection, several open challenges remain to be solved, as we discuss in Section 6.2.

Table 1: Comparison of GLEMOS with previous works providing performances of GNN models.

| | Benchmark Testbeds | Instantaneous Selection Methods | Meta-Graph Features | Graph Learning Models | # Graph Datasets | Graph Size (max # nodes) | # Data Domains |
|---|---|---|---|---|---|---|---|
| GNN-Bank-101 [5] | ✗ | ✗ | ✗ | GNNs | 12 | 34k | 5 |
| NAS-Bench-Graph [32] | ✗ | ✗ | ✗ | GNNs | 9 | 170k | 4 |
| GraphGym [46] | ✗ | ✗ | ✗ | GNNs | 32 | 34k | 7 |
| **GLEMOS (Ours)** | ✓ | ✓ | ✓ | **GNNs & Non-GNNs** (*e.g.*, node2vec, label prop.) | **457** | **496k** | **37** |

Table 2: Summary statistics of the GLEMOS benchmark.

| | Node Classification Task | Link Prediction Task |
|---|---|---|
| **Total performance evaluations** (Section 3) (# model performances on benchmark graphs) | 41,856 | 152,070 |
| **Total graphs** (Sections 3.2 and 3.3) | 128 | 457 |
| ▪ **Num nodes** | 34–421,961 | 34–495,957 |
| ▪ **Num edges** | 156–7,045,181 | 156–7,045,181 |
| ▪ **Num node feats** | 2–61,278 | 2–61,278 |
| ▪ **Num node classes** | 2–195 | N/A |
| ▪ **Num graph domains** | 25 | 37 |
| **Total GL models** (Sections 3.2 and 3.3) | 327 | 350 |
| **Total meta-graph features** (Section 4) | 58–1,074 | 58–1,074 |
| **Total model selection methods** (Section 5) | 10 | 10 |
| **Total benchmark testbeds** (Section 5) | 5 | 5 |

## 2.2 Benchmarks for Instantaneous Graph Learning Model Selection

A few recent works [5, 32, 46] address problems related to GL model selection, and provide performances of GNNs on different datasets. However, all of them perform evaluation-based model selection discussed above, which requires multiple rounds of model evaluations given a new dataset.

As the first benchmark for instantaneous GL model selection (Prob. 1), GLEMOS provides more than just a collection of performance records, *i.e.*, (1) benchmark testbeds and (2) existing algorithms (Section 5) for instantaneous model selection, as well as different sets of (3) meta-graph features (Section 4). These features (1)-(3) are not provided by these previous works [5, 32, 46]. Furthermore, GLEMOS provides more comprehensive and diverse performance records than these works in several aspects (*e.g.*, in terms of included GL models and graph data distributions), as summarized in Table 1.

The two major components for instantaneous GL model selection are *historical model performances* of the GL task of interest (*e.g.*, accuracy for node classification), and *meta-graph features* to quantify graph similarities. For each component, GLEMOS provides several options to choose from. Once these components are chosen, users select a *model selection algorithm*, as well as a *benchmark testbed* to perform evaluation, out of several options available in GLEMOS. Fig. 2 summarizes these steps to use GLEMOS. In the next sections, we describe what GLEMOS provides for these steps.

## 3 Graph Learning Tasks and Performance Collection

Prior model performances play an essential role in instantaneous GL model selection algorithms, as they can estimate a candidate GL model's performance on the new graph based on its observed performances on similar graphs. GLEMOS provides performance collections for two fundamental graph learning tasks, *i.e.*, node classification and link prediction. Below we discuss how the graphs and models are selected, and describe how model performances are evaluated for each GL task.

### 3.1 Graphs and Models

**Graphs.** Our principle of selecting the graphs in GLEMOS is to include diverse graph datasets, in terms of both the size and domain of the graph. The size of selected graphs ranges from a few hundred edges to millions of edges, and the graph set covers various domains, *e.g.*, co-purchase networks, protein networks, citation graphs, and road networks. As listed in Table 1, the resulting graph set outperforms existing data banks in terms of the number and size of graphs, as well as the diversity of data domain. Table 2 shows the summary statistics of graphs, and the graph list is given in Appendix.

**Models.** Our principle for selecting the models to include in GLEMOS is to cover representative and widely-used GL methods. We include graph neural network methods (*e.g.*, GCN [19], GAT [37], and SGC [41]), random walk-based node embeddings (*e.g.*, node2vec [13]), self-supervised graph representation learning methods (*e.g.*, DGI [38]), and classical methods (*e.g.*, spectral embedding [25]). The resulting model set is more diverse than previous works, which considered GNNs alone (Table 1).

### 3.2 Node Classification

**Graph Set.** A subset of the graphs have node labels. Excluding the graphs without node labels, the node classification graph set is comprised of 128 graphs from 25 domains.

Table 3: Graph learning methods and their hyperparameter settings that comprise the model set $\mathcal{M}$. NC: Applicable for node classification. LP: Applicable for link prediction.

| Method | NC | LP | Hyperparameter Settings | Count |
|---|---|---|---|---|
| GCN [19] | ✓ | ✓ | act $a \in \{$relu, tanh, elu$\}$, dropout $d \in \{0.0, 0.5\}$, hidden channels $h \in \{16, 64\}$, num layers $\ell \in \{1, 2, 3\}$ | 30 |
| GraphSAGE [15] | ✓ | ✓ | act $a \in \{$relu, tanh$\}$, aggr $g \in \{$mean, max$\}$, hidden channels $h \in \{16, 64\}$, jumping knowledge $j \in \{$none, last$\}$, num layers $\ell \in \{1, 2\}$ | 24 |
| GAT [37] | ✓ | ✓ | concat $c \in \{$true, false$\}$, dropout $d \in \{0.0, 0.5\}$, heads $n \in \{1, 4\}$, hidden channels $h \in \{16, 64\}$, num layers $\ell \in \{1, 2, 3\}$ | 40 |
| GIN [44] | ✓ | ✓ | eps $e \in \{0.0\}$, hidden channels $h \in \{16, 64\}$, num layers $\ell \in \{1, 2, 3\}$, train eps $t \in \{$true, false$\}$ | 10 |
| EGC [35] | ✓ | ✓ | aggregators $a \in \{$[sum], [mean], [symnorm], [min], [max], [var], [std]$\}$, hidden channels $h \in \{16, 64\}$, num bases $b \in \{4, 8\}$, num layers $\ell \in \{2\}$ | 28 |
| SGC [41] | ✓ | ✓ | bias $b \in \{$true, false$\}$, num hops $k \in \{1, 2, 3, 4, 5\}$ | 10 |
| ChebNet [8] | ✓ | ✓ | Chebyshev filter size $k \in \{1, 2, 3\}$, hidden channels $h \in \{16, 64\}$, normalization $r \in \{$none, sym, rw$\}$, num layers $\ell \in \{1, 2\}$ | 27 |
| PNA [7] | ✓ | ✓ | aggregators $a \in \{$[sum], [mean], [max], [var]$\}$, hidden channels $h \in \{16\}$, num layers $\ell \in \{1, 2\}$, scalers $s \in \{$[identity], [amplification], [attenuation], [linear]$\}$, towers $t \in \{1\}$ | 32 |
| Spectral Emb. [25] | ✓ | ✓ | num components $h \in \{16, 64\}$, tolerance $t \in \{0.1, 0.01, 0.001, 0.0001\}$ | 8 |
| GraRep [6] | ✓ | ✓ | num components $h \in \{16, 32, 64\}$, power $p \in \{1, 2\}$ | 6 |
| DGI [38] | ✓ | ✓ | encoder act $a \in \{$prelu, relu, tanh$\}$, hidden channels $h \in \{16, 64\}$, summary $s \in \{$mean, max, min, var$\}$ | 24 |
| node2vec [13] | ✓ | ✓ | context size $w \in \{5, 10\}$, hidden channels $h \in \{16, 64\}$, $p \in \{1, 2, 4\}$, $q \in \{1, 2, 4\}$, walk length $l \in \{10, 20\}$ | 72 |
| Label Prop. [53] | ✓ | | alpha $\alpha \in \{0.99, 0.9, 0.8, 0.7\}$, num layers $\ell \in \{1, 2, 3, 4\}$ | 16 |
| Jaccard's Coeff. [24] | | ✓ | - | 1 |
| Resource Alloc. [51] | | ✓ | - | 1 |
| Adamic/Adar [24] | | ✓ | - | 1 |
| SEAL [47] | | ✓ | GNN conv $c \in \{$GCN, SAGE, GAT$\}$, GNN hidden channels $g \in \{16, 64, 128\}$, $k \in \{0.6, 0.1\}$, MLP hidden channels $m \in \{32, 128\}$, num hops $n \in \{1\}$ | 36 |
| | | | **Total Count** | **366** |

**Model Set.** Most methods in GLEMOS are applicable for both node classification and link prediction. In addition to these common methods, we also include label propagation [53], which can be used for node classification. The GL models evaluated for node classification and their hyperparameter settings are listed in Table 3. In total, 327 models comprise our model set for node classification.

**Performance Collection.** For node classification, supervised models are optimized to produce the class distribution. For unsupervised models, we first train them to produce latent node embeddings based on their own objective, and apply a trainable linear transform to transform embeddings into the class distribution. More details on the experimental settings are given in the Appendix. To evaluate performance, we calculate multiple classification metrics, including accuracy, F1 score, average precision, and ROC AUC score. Given the graph set $\mathcal{G}$ and model set $\mathcal{M}$ described above, we construct the performance matrix $\mathbf{P}$ by evaluating every model $M_j \in \mathcal{M}$ on every graph $G_i \in \mathcal{G}$, *i.e.*,

$$P_{ij} = \text{Performance } (\textit{e.g.}, \text{ accuracy, and ROC AUC}) \text{ of model } M_j \in \mathcal{M} \text{ on graph } G_i \in \mathcal{G}. \quad (1)$$

*Splitting:* We generate the train-validation-test node splits with a ratio of 64%-16%-20%, respectively, and train each model applying validation-based early stopping. For reproducibility, we release all data splits, such that future model evaluations can be done using the same node splits.

### 3.3 Link Prediction

**Graph Set.** As link prediction task does not require node labels for evaluation, we greatly expand the graph set used for node classification by adding 329 more graphs. With these graphs, the link prediction graph set consists of 457 graphs from 37 domains. The full list is given in the Appendix.

**Model Set.** All models used for node classification are used for link prediction, except for label propagation, which requires node labels. We also add models designed for link prediction, *e.g.*, SEAL [47] and Adamic/Adar [24]. In total, 350 models comprise the link prediction model set(Table 3).

**Performance Collection.** For link prediction, GL models are optimized to produce latent node embeddings, and we apply a dot product scoring between the two node embeddings, followed by a sigmoid function, to obtain the link probability between the corresponding nodes. We calculate multiple evaluation metrics to measure the link prediction performance, including average precision, ROC AUC score, and NDCG (normalized discounted cumulative gain) [39].

*Splitting:* We randomly split edges into train-validation-test sets, with a ratio of 64%-16%-20%, which form positive edge sets. For positive edges, we randomly select the same amount of negative edges (*i.e.*, nonexistent edges), which form negative edge sets. Again, we release all edge splits.

# 4 Collection of Meta-Graph Features

Instantaneous model selection algorithms carry over historical model performances on various graphs to estimate how GL models would perform on a new graph. In that process, performance transfer can be done more effectively when we consider graph similarities, such that the performance transfer would be done adaptively based on the similarities between graphs. Structural meta-graph features provide an effective way to that end by summarizing a graph into a fixed-size feature vector in terms of its structural characteristics. GLEMOS provides various meta-graph features, which can capture important graph structural properties. Below we first discuss how GLEMOS generates fixed-length meta-graph features, as depicted in Figure 3, and then describe the structural features included in GLEMOS, which are organized into three sets for convenience.

**Feature Generation.** GLEMOS produces meta-features in two steps, following an earlier work [30].

○ *Step 1—Structural Feature Extraction:* A structural meta-feature extractor $\psi_k$ is a function that transforms an input graph $G$ into a vector, which represents a distribution of structural features over nodes or edges. For example, degree and PageRank scores of all nodes correspond to node-level feature distributions, and triangle frequency for each edge corresponds to an edge-level feature distribution. In general, we apply a set of such extractors $\Psi = \{\psi_1, \ldots, \psi_K\}$ to the graph $G$, obtaining a set $\Psi(G) = \{\psi_1(G), \ldots, \psi_K(G)\}$ of multiple feature distributions.

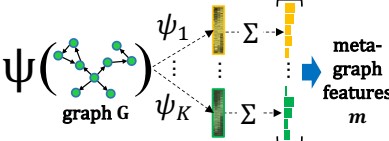

Figure 3: Meta-graph features summarize structural graph characteristics into a fixed-size feature vector.

○ *Step 2—Statistical Feature Summarization:* Since the number of nodes or edges in each graph determines the size of the output from the meta-feature extractors $\psi_k(G)$, those structural feature distributions cannot be directly used to compare graphs with different number of nodes or edges. Step 2 addresses this issue via statistical feature summarization, which applies a set $\Sigma$ of statistical functions (*e.g.*, mean, entropy, skewness, etc) that summarize feature distributions $\psi_k(G)$ of varying size into fixed-length feature vectors; *i.e.*, $\dim(\Sigma(\psi_k(G_i))) = \dim(\Sigma(\psi_k(G_j)))$ for two graphs $G_i$ and $G_j$. By combining all $K$ summaries, graph $G$'s meta-graph feature $\mathbf{m}$ is obtained to be $\mathbf{m} = [\Sigma(\psi_1(G)); \cdots ; \Sigma(\psi_K(G))]$. The statistical functions $\Sigma$ used in GLEMOS are listed in Appendix.

**Collection of Meta-Graph Features.** Different graph features may capture different structural properties. Thus, GLEMOS aims to provide representative and diverse graph features, which have been proven effective in earlier studies, while making it easy to work with any set of features. For the convenience of the users, we group the currently supported features into the following three sets: $\textbf{\textit{M}}_{regular}$ includes widely used features that capture structural characteristics of a graph at both node and graph levels; $\textbf{\textit{M}}_{graphlets}$ considers features based on the frequency of graphlets, as they can provide additional information; $\textbf{\textit{M}}_{compact}$ is intended to use the least space, while providing several important features that capture node-, edge-, and graph-level characteristics. The details of each set are as follows.

$\textbf{\textit{M}}_{regular}$: This set includes 318 meta-graph features. We derive the distribution of node degrees, k-core numbers, PageRank scores, along with the distribution of 3-node paths (wedges) and 3-node cliques (triangles). Given these five distributions, we summarize each using the set of 63 statistical functions $\Sigma$, giving us a total of 315 features. We include three additional features based on the density of graph $G$ and the density of the symmetrized graph, along with the assortativity coefficient.

$\textbf{\textit{M}}_{graphlets}$: This set includes 756 meta-graph features. First, we derive the frequency of all 3 and 4-node graphlet orbits per edge in $G$. Next, we summarize each of the 12 graphlet orbit frequency distributions using the set of 63 statistical functions $\Sigma$, giving us a total of 756 meta-graph features.

$\textbf{\textit{M}}_{compact}$: This set consists of 58 total meta-graph features, including 9 simple statistics such as number of nodes and edges, density of the graph $G$, max vertex degree, average degree, assortativity coefficient, and the maximum k-core number of $G$, along with the mean and median k-core number of a vertex in $G$. We also include the global clustering coefficient, total triangles, as well as the mean and median number of triangles centered at an edge. We further include the total 4-cliques as well as the mean and median number of 4-cliques centered at an edge. Besides the above 16 features, we also compute the frequency of all 3 and 4-node graphlet orbits per edge, and from these 12 frequency distributions, we derive the mean, median, and max. We also derive the graphlet frequency distribution from the counts of all six 4-node graphlets and include those values directly as features.

Note that the framework is flexible, and users can choose to use any set of features, either a subset of the current features (*e.g.*, to further improve efficiency and use less space), or their superset (*e.g.*, to capture distinct structural characteristics using different features in addressing new tasks).

# 5 Benchmark Testbeds and Algorithms

## 5.1 Benchmark Testbeds

GLEMOS provides multiple benchmark testbeds (*i.e.*, evaluation settings and tasks) designed to assess model selection performance in different usage scenarios. We describe them in detail below.

**Fully-Observed Testbed.** In this setup, model selection algorithms are provided with a full performance matrix $\mathbf{P}$ for the given graph learning task, *i.e.*, without any missing entry in $\mathbf{P}$. Accordingly, this testbed measures model selection performance in the most information-rich setting, where all models in the model set $\mathcal{M}$ have been evaluated on all observed graphs.

*Splitting:* We apply a stratified 5-fold cross validation, *i.e.*, graphs are split into five folds, which are (approximately) of the same size, and balanced in terms of graph domains, and then as each fold (20%) is held out to be used for testing, the other folds (80%) are used for model training. Note that graph splits are used to split the performance matrix $\mathbf{P}$ and meta-graph features $\mathbf{M}$.

**Sparse Testbed.** The performance matrix $\mathbf{P}$ in this setting is sparse and partially observed, *i.e.*, we may only have a few observations for each graph. This setting is important since it can be costly to add a new model to the benchmark, which requires training and evaluating the model multiple times on the graphs in the benchmark. By dealing with model selection using a sparse $\mathbf{P}$, this testbed addresses significant practical considerations, *e.g.*, making it more cost-effective to be able to add new models to the benchmark. Using this testbed, researchers can develop and test specialized algorithms capable of learning from such partially-observed performances. To construct a sparse performance matrix $\mathbf{P}'$, we sample uniformly at random $pm$ values from each row of $\mathbf{P}$, where $p$ is the fraction of values to sample and $m$ is the total number of models. This graph-wise sampling strategy ensures the same number of observations for each graph, which matches the practical motivation that we have a limited budget per graph. For this benchmark, we use different sparsity levels $p \in \{0.1, 0.3, 0.5, 0.7, 0.9\}$.

*Splitting:* We use the same stratified 5-fold cross validation as in Fully-Observed testbed. Algorithms are trained using a sparse split of $\mathbf{P}$, and evaluated with fully-observed performances of the test split.

**Out-Of-Domain Testbed.** Graphs from a particular domain (*e.g.*, road graphs, social networks, and brain networks) often have similar characteristics to each other. In other words, graphs from a certain domain can be considered as having its own distribution, which makes model selection for graphs from a new domain a challenging task. This testbed evaluates the effectiveness of model selection methods for such an out-of-distribution setting by holding out graphs from a specific network domain, and trying to predict for the held-out domain by learning from graphs from all the other domains.

*Splitting:* We use a group-based 5-fold cross validation for this testbed such that each domain appears once in the test set across all folds.

**Small-To-Large Testbed.** Training a GL model can take a lot of time and resources, especially for large-scale graphs. While model selection methods may benefit from having more prior performances, having to obtain performance records for large graphs presents a significant computational bottleneck. The meta-training process can be made significantly faster by enabling model selection algorithms to learn from relatively small graphs to be able to predict for larger graphs. This testbed focuses on this challenging yet practical setting, which evaluates the ability to generalize from small to large graphs.

*Splitting:* Graphs with less than $\epsilon$ nodes form a small-graph set used for training. The other graphs with at least $\epsilon$ nodes form a large-graph set, which is used for evaluation. We evaluate using a threshold value $\epsilon$ of 10000 for this testbed.

**Cross-Task Testbed.** The above testbeds operate on the model performances measured for one specific type of GL task. By contrast, in this testbed, model selection methods learn from performances of one GL task (*e.g.*, node classification), and are evaluated by predicting performances of a different GL task (*e.g.*, link prediction). This task present an additional challenge to model the relation between two different, yet related GL tasks, and utilize the learned knowledge for transferable model selection.

*Splitting:* We first choose the source and target tasks, and split the graphs into the two sets, *i.e.*, the source task set and the target task set. Then the graphs in the source set are used for training, and the graphs in the target set are used for testing.

## 5.2 Model Selection Algorithms

GLEMOS provides state-of-the art algorithms for instantaneous model selection, which are listed in Table 4. These algorithms are selected such that the benchmark covers representative techniques

Table 4: GLEMOS provides representative algorithms for instantaneous model selection. Algorithm characteristics denote whether they utilize meta-graph features (C1) and observed model performances (C2) for model selection, and whether they are optimizable (*i.e.*, have trainable parameters) (C3).

| Characteristics ╲ Algorithm | Random Selection | GB-Avg Perf [49] | GB-Avg Rank [30] | ISAC [18] | AS [27] | Spv. Surro. (S2) [45] | ALORS [26] | NCF [16] | MetaOD [49] | MetaGL [30] |
|---|---|---|---|---|---|---|---|---|---|---|
| C1. Use meta-features | ✗ | ✗ | ✗ | ✓ | ✓ | ✓ | ✓ | ✓ | ✓ | ✓ |
| C2. Use prior performances | ✗ | ✓ | ✓ | ✓ | ✓ | ✓ | ✓ | ✓ | ✓ | ✓ |
| C3. Optimizable | ✗ | ✗ | ✗ | ✗ | ✗ | ✓ | ✓ | ✓ | ✓ | ✓ |

Table 5: Fully-Observed testbed results for link prediction (top) and node classification (bottom) tasks. Higher (↑) scores are better. The **best** result is in bold, and the second best result is underlined.

(a) Link prediction

| Perf. Metric | RandSel | GB-Perf | GB-Rank | ISAC | AS | S2 | ALORS | NCF | MetaOD | MetaGL |
|---|---|---|---|---|---|---|---|---|---|---|
| **AUC** (↑) | 0.524 | 0.735 | 0.730 | 0.807 | 0.864 | 0.809 | 0.843 | 0.728 | 0.764 | **0.875** |
| **MRR** (↑) | 0.016 | 0.087 | 0.064 | 0.134 | **0.371** | 0.198 | 0.201 | 0.073 | 0.096 | 0.295 |
| **NDCG@1** (↑) | 0.813 | 0.942 | 0.934 | 0.944 | 0.957 | 0.950 | 0.961 | 0.943 | 0.937 | **0.969** |

(b) Node classification

| Perf. Metric | RandSel | GB-Perf | GB-Rank | ISAC | AS | S2 | ALORS | NCF | MetaOD | MetaGL |
|---|---|---|---|---|---|---|---|---|---|---|
| **AUC** (↑) | 0.518 | 0.747 | 0.744 | 0.746 | 0.762 | **0.772** | 0.734 | 0.745 | 0.581 | 0.740 |
| **MRR** (↑) | 0.029 | 0.102 | 0.124 | 0.118 | **0.181** | 0.110 | 0.103 | 0.124 | 0.041 | 0.129 |
| **NDCG@1** (↑) | 0.747 | 0.865 | 0.860 | 0.885 | 0.892 | **0.916** | 0.886 | 0.883 | 0.839 | 0.863 |

for model selection, in terms of whether they use meta-graph features (C1, Section 4) and prior model performances (C2, Section 3), and whether they are optimizable with trainable parameters (C3). *Random Selection (RandSel)* is used as a baseline to see how well model selection algorithms perform in comparison to random scoring. *Global Best (GB)-AvgPerf* and *GB-AvgRank* select a model that performed globally well on average. In contrast, *ISAC* [18] and *ARGOSMART (AS)* [27] perform model selection more locally with respect to the given graph, using meta-features. As GB methods rely only on prior performance, comparisons against them can help with investigating the effectiveness of meta-graph features. *Supervised Surrogates (S2)* [45], *ALORS* [26], *NCF* [16], *MetaOD* [49], and *MetaGL* [30] are optimizable algorithms, which learn to estimate model performance by capturing the relation between meta-features and observed performances. In comparison to the simpler, non-optimizable algorithms above, we can investigate the advantages of different optimization components for instantaneous model selection. A more detailed description of each algorithm is given in Appendix.

## 6 Experiments

In this section, we report how model selection methods perform in different testbeds. Based on those observations, we discuss the limitations of existing methods and future research directions.

### 6.1 Model Selection Performance

**Evaluation Protocol.** To measure how well model selection methods perform on the testbeds presented in Section 5, we evaluate their top-1 prediction results (*i.e.*, the model predicted to be the best for the query graph) as model selection aims to find the best performing model as accurately as possible. Specifically, top-1 prediction performance is measured in terms of AUC, MAP (mean average precision), and NDCG (normalized discounted cumulative gain), all of which range from zero to one, with larger values indicating a better performance. We apply AUC and MAP by treating the task as a binary classification problem, in which the top-1 model is labeled as one, and all other models are labeled as zero. For NDCG, we report NDCG@1, which evaluates the ranking quality of the top-1 model. We evaluate these metrics multiple times for the data splits each testbed provides, and report the averaged performance. For reproducibility, GLEMOS provides the data splits of all testbeds.

**Fully-Observed Testbed** (Table 5). Comparison between methods where meta-graph features are either used (*e.g.*, AS, MetaGL) or not used (*e.g.*, GB-Perf) shows the benefits of utilizing meta-graph features for GL model selection. While optimizable methods (*e.g.*, NCF, MetaOD) have the additional flexibility to adaptively tune their behavior based on data, they are outperformed by relatively simple methods like ISAC and AS. At the same time, the best results on link prediction in the majority of metrics are achieved by another optimizable method, MetaGL, which shows the promising potential of optimizable framework for model selection. In node classification results, the performance decrease

Table 6: Sparse testbed results for link prediction (top) and node classification (bottom) tasks. Higher (↑) scores are better. The **best** result is in bold, and the second best result is underlined.

(a) Link prediction

| Perf. Metric | Sparsity | RandSel | GB-Perf | GB-Rank | ISAC | AS | S2 | ALORS | NCF | MetaOD | MetaGL |
|---|---|---|---|---|---|---|---|---|---|---|---|
| **AUC** (↑) | 10% | 0.524 | 0.733 | 0.732 | 0.804 | 0.829 | 0.813 | 0.831 | 0.735 | 0.743 | **0.865** |
| | 30% | 0.524 | 0.728 | 0.738 | 0.798 | 0.763 | 0.811 | 0.827 | 0.739 | 0.703 | **0.871** |
| | 50% | 0.524 | 0.704 | 0.730 | 0.790 | 0.690 | 0.839 | 0.814 | 0.739 | 0.669 | **0.866** |
| | 70% | 0.524 | 0.708 | 0.730 | 0.778 | 0.618 | 0.814 | 0.795 | 0.757 | 0.630 | **0.866** |
| | 90% | 0.524 | 0.717 | 0.732 | 0.720 | 0.547 | 0.464 | 0.687 | 0.656 | 0.599 | **0.811** |

(b) Node classification

| Perf. Metric | Sparsity | RandSel | GB-Perf | GB-Rank | ISAC | AS | S2 | ALORS | NCF | MetaOD | MetaGL |
|---|---|---|---|---|---|---|---|---|---|---|---|
| **AUC** (↑) | 10% | 0.518 | 0.746 | 0.744 | 0.744 | 0.748 | **0.766** | 0.727 | 0.727 | 0.575 | 0.761 |
| | 30% | 0.518 | 0.743 | 0.738 | 0.734 | 0.680 | **0.769** | 0.741 | 0.735 | 0.533 | 0.736 |
| | 50% | 0.518 | 0.726 | **0.739** | 0.687 | 0.592 | 0.739 | 0.730 | 0.713 | 0.485 | 0.709 |
| | 70% | 0.518 | 0.692 | **0.738** | 0.653 | 0.571 | 0.684 | 0.694 | 0.709 | 0.483 | 0.662 |
| | 90% | 0.518 | 0.626 | **0.697** | 0.592 | 0.535 | 0.620 | 0.654 | 0.660 | 0.490 | 0.659 |

Table 7: Out-Of-Domain testbed results for link prediction (top) and node classification (bottom) tasks. Higher (↑) scores are better. The **best** result is in bold, and the second best result is underlined.

(a) Link prediction

| Perf. Metric | RandSel | GB-Perf | GB-Rank | ISAC | AS | S2 | ALORS | NCF | MetaOD | MetaGL |
|---|---|---|---|---|---|---|---|---|---|---|
| **AUC** (↑) | 0.517 | 0.809 | 0.811 | 0.850 | 0.786 | 0.837 | 0.820 | 0.837 | 0.681 | **0.871** |
| **MRR** (↑) | 0.018 | 0.110 | 0.101 | 0.125 | **0.237** | 0.116 | 0.109 | 0.109 | 0.047 | 0.148 |
| **NDCG@1** (↑) | 0.820 | **0.956** | 0.954 | 0.951 | 0.935 | 0.953 | 0.953 | 0.952 | 0.918 | 0.951 |

(b) Node classification

| Perf. Metric | RandSel | GB-Perf | GB-Rank | ISAC | AS | S2 | ALORS | NCF | MetaOD | MetaGL |
|---|---|---|---|---|---|---|---|---|---|---|
| **AUC** (↑) | 0.495 | 0.726 | 0.727 | 0.701 | 0.684 | **0.750** | 0.668 | 0.741 | 0.571 | 0.705 |
| **MRR** (↑) | 0.019 | 0.074 | 0.086 | 0.046 | 0.060 | **0.089** | 0.056 | 0.066 | 0.044 | 0.082 |
| **NDCG@1** (↑) | 0.722 | 0.828 | 0.836 | 0.848 | 0.828 | **0.901** | 0.796 | 0.842 | 0.810 | 0.848 |

of optimizable methods are notable (*e.g.*, MetaGL). One potential reason for this is that the graph set for node classification is relatively small compared to the graphs applicable for link prediction, which limits the effectiveness of optimizable algorithms that are more prone to overfitting in such cases.

**Sparse Testbed** (Table 6). As the sparsity of the performance matrix **P** increases, model selection methods perform increasingly worse. In particular, while AS achieves the best or the second best results in the Fully-Observed testbed, its performance quickly declines as sparsity increases. Since AS performs model selection based on the most similar observed graph, it cannot operate effectively in a highly sparse setting. Global averaging methods (*e.g.*, GB-Perf), or more sophisticated optimizable methods show more stable results. Due to the additional requirement for node labels, node classification task in this setup presents the most data sparse, yet practically important regime.

**Out-Of-Domain Testbed** (Table 7). Graphs in the same or similar domains are often more similar to each other than graphs in different domains. As this testbed requires addressing additional challenges to achieve out-of-distribution generalization, most methods perform worse than in other in-distribution testbeds. For instance, AS, which are sensitive to the availability of observed graphs similar to the query graph, perform worse than in Table 5. On the other hand, optimizable methods show more promising results as they learn to extrapolate into new domains by learning from observed domains.

**Small-To-Large Testbed** (Table 8). In comparison to the Fully-Observed testbed, the performance decreases overall in this testbed. However, considering that methods learn only from small graphs, model selection for large graphs still performs quite well, often achieving a similar level of performance. Successful methods in this testbed can make the model selection pipeline much more efficient as performance collection for small graphs can be done much more efficiently than for large graphs.

**Additional Results.** We provide additional results in the Appendix, including the results of the Cross-Task testbed, and results obtained with other meta-graph features, *e.g.*, $M_{graphlets}$ and $M_{compact}$.

Table 8: Small-To-Large testbed results for link prediction (top) and node classification (bottom) tasks. Higher (↑) scores are better. The **best** result is in bold, and the second best result is underlined.

(a) Link prediction

| Perf. Metric | RandSel | GB-Perf | GB-Rank | ISAC | AS | S2 | ALORS | NCF | MetaOD | MetaGL |
|---|---|---|---|---|---|---|---|---|---|---|
| **AUC** (↑) | 0.522 | 0.798 | 0.797 | 0.842 | 0.827 | 0.767 | 0.812 | 0.796 | 0.667 | **0.870** |
| **MRR** (↑) | 0.031 | 0.072 | 0.061 | 0.132 | **0.368** | 0.074 | 0.209 | 0.047 | 0.075 | 0.260 |
| **NDCG@1** (↑) | 0.841 | 0.958 | 0.960 | 0.957 | 0.951 | 0.953 | 0.947 | 0.956 | 0.921 | **0.964** |

(b) Node classification

| Perf. Metric | RandSel | GB-Perf | GB-Rank | ISAC | AS | S2 | ALORS | NCF | MetaOD | MetaGL |
|---|---|---|---|---|---|---|---|---|---|---|
| **AUC** (↑) | 0.508 | 0.724 | 0.726 | 0.711 | **0.761** | 0.664 | 0.701 | 0.697 | 0.467 | 0.736 |
| **MRR** (↑) | 0.011 | 0.058 | 0.082 | **0.109** | 0.095 | 0.036 | 0.042 | 0.034 | 0.016 | 0.071 |
| **NDCG@1** (↑) | 0.795 | 0.861 | 0.896 | **0.902** | 0.883 | 0.855 | 0.862 | 0.864 | 0.830 | 0.857 |

## 6.2 Discussion on Limitations and Future Directions

**Limitations.** In principle, using GLEMOS to select a GL model to employ for a new graph is based on the assumption that similar graph datasets exist in the benchmark. Therefore, it may not be very effective if the new graph is significantly different from all graphs in the benchmark (*e.g.*, the new graph is from a completely new domain). However, as the benchmark data continue to grow over time, such cases will be increasingly less likely, while model selection performances will likely improve with the addition of more data. Furthermore, while GLEMOS currently supports two fundamental GL tasks, namely, node classification and link prediction, it can be further extended with additional tasks (*e.g.*, graph classification). Incorporating them into GLEMOS is one of the future plans.

**Future Directions.** Below we list promising research directions to further improve the algorithms as well as the benchmark for instantaneous GL model selection.

▪ *Direction 1: enabling model selection methods to use additional graph data.* While existing methods utilize model performances and graph structural information captured by meta-features, they currently do not take other available graph data into account, such as node and edge features, timestamps in the case of dynamic graphs, and node and edge types (*e.g.*, knowledge graphs). These data can be useful for modeling graph similarities, and the benchmark can further be enriched with such additional data.

▪ *Direction 2: developing data augmentation techniques.* Adding new performance records to the benchmark can improve the effectiveness of model selection methods. However, it is often computationally expensive to train and evaluate GL models on non-trivial graphs. Data augmentation techniques for GL model performances can be helpful in this data sparse regime, especially for optimizable methods that require a lot of data to learn effectively.

▪ *Direction 3: handling out-of-distribution settings.* Existing model selection methods are mainly designed for an in-distribution setup, as they assume that there exist observed graphs similar to a query graph. Thus their performance is suboptimal when a query graph comes from a new distribution. Investigating how to achieve generalization in such an out-of-distribution scenario would be beneficial.

▪ *Direction 4: effective performance collection.* When we have a limited budget for performance measurements on new graphs, selecting which pairs of graphs and models to evaluate and include in the benchmark can greatly influence the learning of model selection methods. Thus the ability to find a small set of representative pairs can lead to a fast and effective performance collection. Challenges include how to make such selections from a heterogeneous model set with multiple GL methods.

## 7 Conclusion

The choice of a GL model has a significant impact on the performance of downstream tasks. Despite recent efforts to tackle this important problem, there exists no benchmark environment to evaluate the performance of GL model selection methods, and to support the development of new methods. In this work, we develop GLEMOS, the first benchmark environment for instantaneous GL model selection.

- **Extensive Benchmark Data.** Among others, GLEMOS provides an extensive collection of model performances on fundamental GL tasks, *i.e.*, link prediction and node classification, which is by far the largest and most comprehensive benchmark for Prob. 1 to the best of our knowledge.
- **Algorithms and Testbeds.** GLEMOS provides representative algorithms for Prob. 1, as well as multiple testbeds to assess model selection performance in practical usage scenarios.
- **Extensible Open Source Environment.** GLEMOS is designed to be easily extended with new GL models, new graphs, new performance records, and new GL tasks, while allowing reproducibility.

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
