# Appendix to "GLEMOS: Benchmark for Instantaneous Graph Learning Model Selection"

**Namyong Park**[1]*, **Ryan Rossi**[2], **Xing Wang**[1], **Antoine Simoulin**[1],
**Nesreen Ahmed**[3], **Christos Faloutsos**[4]
[1]Meta AI [2]Adobe Research [3]Intel Labs [4]Carnegie Mellon University

## Abstract

In this appendix, we provide additional experimental results; details of model selection algorithms, and meta-graph features; experimental settings; a discussion on the usage and extensibility of GLEMOS; details of the data GLEMOS provides; and hosting, licensing, and maintenance plan.

**Access to GLEMOS.** The code and data of GLEMOS can be accessed from `https://namyongpark.github.io/glemos`.

## A  Additional Results

### A.1  Cross-Task Testbed Results

In Table 1, we report the cross-task testbed results in two transfer learning settings, *i.e.*, (a) node classification to link prediction (Table 1a) and (b) link prediction to node classification (Table 1b). Compared to other testbeds that operate on the performances measured for only one type of GL task, nearly all methods exhibit performance decline in this challenging setup, which indicates that GL models that are good for one type of task may not be as effective for another type of GL task. In contrast to other testbeds, sophisticated algorithms (*e.g.*, MetaGL) tend to experience more performance decrease in this testbed than simple averaging methods (*e.g.*, GB-Rank), which perform close to the best. By designing mechanisms that can model how performance characteristics on one task would translate to those on another, optimizable algorithms can be made much more effective in this setting.

Table 1: Cross-Task testbed results for node classification-to-link prediction (top) and link prediction-to-node classification (bottom) settings. Higher (↑) scores are better. The numbers in the parentheses denote one standard error. The **best** result is in bold, and the second best result is underlined.

(a) Node classification → Link prediction

| Perf. Metric | RandSel | GB-Perf | GB-Rank | ISAC | AS | S2 | ALORS | NCF | MetaOD | MetaGL |
|---|---|---|---|---|---|---|---|---|---|---|
| **AUC** (↑) | 0.479 (0.016) | 0.652 (0.013) | **0.671** (0.013) | 0.626 (0.014) | 0.520 (0.015) | 0.636 (0.014) | 0.594 (0.015) | 0.650 (0.013) | 0.465 (0.015) | 0.553 (0.014) |
| **MRR** (↑) | 0.020 (0.003) | 0.022 (0.002) | 0.022 (0.002) | 0.020 (0.002) | 0.026 (0.006) | 0.022 (0.002) | **0.029** (0.005) | 0.024 (0.002) | 0.015 (0.003) | 0.018 (0.002) |
| **NDCG@1** (↑) | 0.830 (0.008) | 0.806 (0.008) | 0.848 (0.006) | 0.833 (0.007) | 0.825 (0.008) | 0.815 (0.007) | **0.858** (0.007) | 0.812 (0.008) | 0.807 (0.008) | 0.827 (0.008) |

(b) Link prediction → Node classification

| Perf. Metric | RandSel | GB-Perf | GB-Rank | ISAC | AS | S2 | ALORS | NCF | MetaOD | MetaGL |
|---|---|---|---|---|---|---|---|---|---|---|
| **AUC** (↑) | 0.542 (0.025) | 0.595 (0.025) | 0.596 (0.025) | 0.581 (0.026) | 0.517 (0.025) | 0.536 (0.024) | 0.606 (0.025) | **0.608** (0.025) | 0.477 (0.024) | 0.542 (0.026) |
| **MRR** (↑) | 0.023 (0.008) | 0.053 (0.016) | 0.047 (0.011) | 0.043 (0.012) | 0.027 (0.009) | 0.034 (0.010) | 0.032 (0.006) | **0.057** (0.016) | 0.011 (0.001) | 0.038 (0.012) |
| **NDCG@1** (↑) | 0.731 (0.022) | 0.817 (0.018) | 0.822 (0.019) | 0.795 (0.021) | 0.780 (0.020) | 0.789 (0.021) | 0.747 (0.022) | **0.823** (0.017) | 0.730 (0.021) | 0.779 (0.021) |

---

*Correspondence: namyongp@meta.com

37th Conference on Neural Information Processing Systems (NeurIPS 2023) Track on Datasets and Benchmarks.

## A.2 Results With Different Sets of Meta-Graph Features

We present three sets of meta-graph features in the main text, *i.e.*, $M_{regular}$, $M_{graphlets}$, and $M_{compact}$, which consist of different types of graph structural features. Here we provide these different sets of meta-graph features to model selection algorithms, and evaluate their performance using each set. In addition to the above three sets, we also use the concatenation of $M_{regular}$ and $M_{graphlets}$, denoted $M_{reg+graphlets}$, which augments the regular structural features with graphlet-based features, forming the largest set with 1074 meta-graph features. In Tables 2 to 6, we report the performance of model selection algorithms in terms of their ROC AUC scores for the five testbeds. Since the Random Selection and Global Best (GB) algorithms are independent of meta-features, their performances are the same across different features. From the results below, we make the following observations.

***Using additional meta-graph features can improve model selection results.*** For instance, in Sparse testbed (Table 3), the best performing method, MetaGL, achieves the highest AUC by using an augmented feature set $M_{reg+graphlets}$. As distinct graph features may capture different aspects of graph structural properties, they can provide further information to find a better model.

***More features do not always lead to a better performance.*** For example, in Tables 2 and 3, we see mixed results with optimizable methods (*e.g.*, NCF). In some cases, they experience some performance improvements, while in others, their performance declines as they use more features. The capability to adaptively utilize meta-features for the given context could further improve their performances.

***The impact of different meta-graph features is more pronounced in the more challenging transfer learning settings***, *i.e.*, Out-Of-Domain (Table 4), Small-To-Large (Table 5), and Cross-Task (Table 6) testbeds. These testbeds present additional challenges for model selection methods to achieve an effective generalization (*e.g.*, large differences exist in graph data distributions or graph sizes between training and testing phases). As existing methods do not take such challenges into account, they are prone to overfitting and thus may not generalize well in the testing phase. For example, the performance of MetaGL in the Cross-Task testbed (Table 6) is the lowest when using the largest feature set $M_{reg+graphlets}$. A stronger and more robust transfer capability would be needed to enable a better use of additional meta-graph features in such cases. On the other hand, we also observe that using more meta-features leads to a significant performance improvement for relatively simple methods, *e.g.*, ISAC in Small-To-Large testbed (Table 5), which shows the promises and potential of meta-graph features to handle these challenging transfer learning settings.

Table 2: Fully-Observed testbed results, obtained with different meta-graph features, for link prediction (top) and node classification (bottom) tasks. $M_{reg+graphlets}$ denotes a concatenation of $M_{regular}$ and $M_{graphlets}$ meta-graph features. Higher (↑) scores are better. The numbers in the parentheses denote one standard error. The **best** result is in bold, and the second best result is underlined.

(a) Link prediction

| Perf. Metric | Meta-Feature | RandSel | GB-Perf | GB-Rank | ISAC | AS | S2 | ALORS | NCF | MetaOD | MetaGL |
|---|---|---|---|---|---|---|---|---|---|---|---|
| **AUC (↑)** | $M_{compact}$ | 0.524 (0.013) | 0.735 (0.011) | 0.730 (0.010) | 0.757 (0.011) | 0.870 (0.010) | 0.831 (0.009) | 0.847 (0.009) | 0.789 (0.011) | 0.726 (0.015) | **0.875** (0.009) |
| | $M_{regular}$ | 0.524 (0.013) | 0.735 (0.011) | 0.730 (0.010) | 0.807 (0.011) | 0.864 (0.010) | 0.809 (0.011) | 0.843 (0.010) | 0.728 (0.011) | 0.764 (0.014) | **0.875** (0.009) |
| | $M_{graphlets}$ | 0.524 (0.013) | 0.735 (0.011) | 0.730 (0.010) | 0.781 (0.011) | 0.850 (0.011) | 0.806 (0.011) | 0.830 (0.010) | 0.791 (0.011) | 0.740 (0.014) | **0.873** (0.009) |
| | $M_{reg+graphlets}$ | 0.524 (0.013) | 0.735 (0.011) | 0.730 (0.010) | 0.803 (0.011) | 0.867 (0.010) | 0.833 (0.010) | 0.843 (0.010) | 0.794 (0.010) | 0.740 (0.014) | **0.874** (0.009) |

(b) Node classification

| Perf. Metric | Meta-Feature | RandSel | GB-Perf | GB-Rank | ISAC | AS | S2 | ALORS | NCF | MetaOD | MetaGL |
|---|---|---|---|---|---|---|---|---|---|---|---|
| **AUC (↑)** | $M_{compact}$ | 0.518 (0.026) | 0.747 (0.024) | 0.744 (0.024) | 0.749 (0.024) | **0.786** (0.023) | 0.775 (0.022) | 0.763 (0.023) | 0.744 (0.024) | 0.602 (0.029) | 0.765 (0.023) |
| | $M_{regular}$ | 0.518 (0.026) | 0.747 (0.024) | 0.744 (0.024) | 0.746 (0.023) | 0.762 (0.023) | **0.772** (0.022) | 0.734 (0.023) | 0.745 (0.025) | 0.581 (0.028) | 0.740 (0.024) |
| | $M_{graphlets}$ | 0.518 (0.026) | 0.747 (0.024) | 0.744 (0.024) | 0.746 (0.024) | 0.729 (0.026) | 0.747 (0.023) | 0.715 (0.025) | 0.743 (0.024) | 0.629 (0.029) | **0.763** (0.023) |
| | $M_{reg+graphlets}$ | 0.518 (0.026) | 0.747 (0.024) | 0.744 (0.024) | **0.758** (0.023) | 0.728 (0.026) | 0.742 (0.025) | 0.744 (0.023) | 0.735 (0.025) | 0.600 (0.029) | 0.734 (0.025) |

Table 3: Sparse testbed results, obtained with different meta-graph features, and performance matrices with a sparsity of $50\%$, for link prediction (top) and node classification (bottom) tasks. $M_{reg+graphlets}$ denotes a concatenation of $M_{regular}$ and $M_{graphlets}$ meta-graph features. Higher ($\uparrow$) scores are better. The numbers in the parentheses denote one standard error. The **best** result is in bold, and the second best result is underlined.

(a) Link prediction

| Perf. Metric | Meta-Feature | RandSel | GB-Perf | GB-Rank | ISAC | AS | S2 | ALORS | NCF | MetaOD | MetaGL |
|---|---|---|---|---|---|---|---|---|---|---|---|
| AUC ($\uparrow$) | $M_{compact}$ | 0.524 (0.013) | 0.704 (0.011) | 0.730 (0.010) | 0.741 (0.011) | 0.682 (0.012) | 0.829 (0.009) | 0.802 (0.010) | 0.780 (0.011) | 0.678 (0.014) | **0.865** (0.010) |
| | $M_{regular}$ | 0.524 (0.013) | 0.704 (0.011) | 0.730 (0.010) | 0.790 (0.011) | 0.690 (0.012) | 0.839 (0.010) | 0.814 (0.010) | 0.739 (0.011) | 0.669 (0.015) | **0.866** (0.010) |
| | $M_{graphlets}$ | 0.524 (0.013) | 0.704 (0.011) | 0.730 (0.010) | 0.762 (0.011) | 0.690 (0.013) | 0.814 (0.010) | 0.802 (0.010) | 0.775 (0.011) | 0.646 (0.015) | **0.871** (0.009) |
| | $M_{reg+graphlets}$ | 0.524 (0.013) | 0.704 (0.011) | 0.730 (0.010) | 0.787 (0.011) | 0.679 (0.012) | 0.838 (0.010) | 0.808 (0.010) | 0.778 (0.010) | 0.676 (0.014) | **0.875** (0.010) |

(b) Node classification

| Perf. Metric | Meta-Feature | RandSel | GB-Perf | GB-Rank | ISAC | AS | S2 | ALORS | NCF | MetaOD | MetaGL |
|---|---|---|---|---|---|---|---|---|---|---|---|
| AUC ($\uparrow$) | $M_{compact}$ | 0.518 (0.026) | 0.726 (0.024) | 0.739 (0.024) | 0.694 (0.025) | 0.652 (0.023) | 0.727 (0.023) | 0.731 (0.023) | 0.731 (0.024) | 0.471 (0.030) | **0.748** (0.024) |
| | $M_{regular}$ | 0.518 (0.026) | 0.726 (0.024) | **0.739** (0.024) | 0.687 (0.022) | 0.592 (0.021) | 0.739 (0.024) | 0.730 (0.024) | 0.713 (0.025) | 0.485 (0.031) | 0.709 (0.023) |
| | $M_{graphlets}$ | 0.518 (0.026) | 0.726 (0.024) | 0.739 (0.024) | 0.687 (0.024) | 0.591 (0.024) | 0.682 (0.025) | **0.751** (0.023) | 0.721 (0.023) | 0.497 (0.030) | 0.731 (0.025) |
| | $M_{reg+graphlets}$ | 0.518 (0.026) | 0.726 (0.024) | **0.739** (0.024) | 0.694 (0.024) | 0.606 (0.025) | 0.721 (0.024) | 0.731 (0.022) | 0.705 (0.024) | 0.494 (0.029) | 0.711 (0.023) |

Table 4: Out-Of-Domain testbed results, obtained with different meta-graph features, for link prediction (top) and node classification (bottom) tasks. $M_{reg+graphlets}$ denotes a concatenation of $M_{regular}$ and $M_{graphlets}$ meta-graph features. Higher ($\uparrow$) scores are better. The numbers in the parentheses denote one standard error. The **best** result is in bold, and the second best result is underlined.

(a) Link prediction

| Perf. Metric | Meta-Feature | RandSel | GB-Perf | GB-Rank | ISAC | AS | S2 | ALORS | NCF | MetaOD | MetaGL |
|---|---|---|---|---|---|---|---|---|---|---|---|
| AUC ($\uparrow$) | $M_{compact}$ | 0.517 (0.013) | 0.809 (0.010) | 0.811 (0.010) | 0.805 (0.012) | 0.779 (0.012) | 0.813 (0.010) | 0.832 (0.009) | 0.816 (0.010) | 0.659 (0.015) | **0.867** (0.009) |
| | $M_{regular}$ | 0.517 (0.013) | 0.809 (0.010) | 0.811 (0.010) | 0.850 (0.009) | 0.786 (0.012) | 0.837 (0.010) | 0.820 (0.009) | 0.837 (0.009) | 0.681 (0.015) | **0.871** (0.009) |
| | $M_{graphlets}$ | 0.517 (0.013) | 0.809 (0.010) | 0.811 (0.010) | 0.828 (0.011) | 0.788 (0.011) | 0.777 (0.012) | 0.807 (0.010) | 0.828 (0.010) | 0.689 (0.015) | **0.864** (0.009) |
| | $M_{reg+graphlets}$ | 0.517 (0.013) | 0.809 (0.010) | 0.811 (0.010) | 0.843 (0.011) | 0.790 (0.011) | 0.839 (0.009) | 0.830 (0.009) | 0.832 (0.010) | 0.688 (0.015) | **0.864** (0.009) |

(b) Node classification

| Perf. Metric | Meta-Feature | RandSel | GB-Perf | GB-Rank | ISAC | AS | S2 | ALORS | NCF | MetaOD | MetaGL |
|---|---|---|---|---|---|---|---|---|---|---|---|
| AUC ($\uparrow$) | $M_{compact}$ | 0.495 (0.025) | 0.726 (0.025) | 0.727 (0.025) | 0.718 (0.025) | 0.646 (0.025) | 0.728 (0.022) | 0.690 (0.023) | **0.736** (0.023) | 0.468 (0.027) | 0.715 (0.024) |
| | $M_{regular}$ | 0.495 (0.025) | 0.726 (0.025) | 0.727 (0.025) | 0.701 (0.025) | 0.684 (0.023) | **0.750** (0.023) | 0.668 (0.026) | 0.741 (0.024) | 0.571 (0.027) | 0.705 (0.023) |
| | $M_{graphlets}$ | 0.495 (0.025) | 0.726 (0.025) | 0.727 (0.025) | 0.697 (0.025) | 0.673 (0.026) | **0.746** (0.022) | 0.677 (0.024) | 0.717 (0.024) | 0.537 (0.030) | 0.688 (0.024) |
| | $M_{reg+graphlets}$ | 0.495 (0.025) | 0.726 (0.025) | 0.727 (0.025) | 0.691 (0.024) | 0.667 (0.025) | **0.732** (0.023) | 0.628 (0.025) | 0.712 (0.026) | 0.536 (0.029) | 0.660 (0.025) |

Table 5: Small-To-Large testbed results, obtained with different meta-graph features, for link prediction (top) and node classification (bottom) tasks. $M_{reg+graphlets}$ denotes a concatenation of $M_{regular}$ and $M_{graphlets}$ meta-graph features. Higher (↑) scores are better. The numbers in the parentheses denote one standard error. The **best** result is in bold, and the second best result is underlined.

(a) Link prediction

| Perf. Metric | Meta-Feature | RandSel | GB-Perf | GB-Rank | ISAC | AS | S2 | ALORS | NCF | MetaOD | MetaGL |
|---|---|---|---|---|---|---|---|---|---|---|---|
| | $M_{compact}$ | 0.522 (0.027) | 0.798 (0.017) | 0.797 (0.017) | 0.772 (0.022) | 0.835 (0.020) | 0.715 (0.021) | 0.837 (0.018) | 0.783 (0.017) | 0.480 (0.029) | **0.875** (0.018) |
| | $M_{regular}$ | 0.522 (0.027) | 0.798 (0.017) | 0.797 (0.017) | 0.842 (0.018) | 0.827 (0.022) | 0.767 (0.020) | 0.812 (0.024) | 0.796 (0.017) | 0.667 (0.029) | **0.870** (0.018) |
| AUC (↑) | $M_{graphlets}$ | 0.522 (0.027) | 0.798 (0.017) | 0.797 (0.017) | 0.841 (0.016) | 0.830 (0.019) | 0.750 (0.021) | 0.842 (0.020) | 0.783 (0.018) | 0.700 (0.028) | **0.875** (0.016) |
| | $M_{reg+graphlets}$ | 0.522 (0.027) | 0.798 (0.017) | 0.797 (0.017) | 0.843 (0.016) | 0.841 (0.020) | 0.806 (0.018) | 0.831 (0.021) | 0.795 (0.018) | 0.750 (0.027) | **0.858** (0.018) |

(b) Node classification

| Perf. Metric | Meta-Feature | RandSel | GB-Perf | GB-Rank | ISAC | AS | S2 | ALORS | NCF | MetaOD | MetaGL |
|---|---|---|---|---|---|---|---|---|---|---|---|
| | $M_{compact}$ | 0.508 (0.039) | 0.724 (0.042) | **0.726** (0.042) | 0.655 (0.043) | 0.723 (0.038) | 0.682 (0.044) | 0.696 (0.041) | 0.693 (0.039) | 0.586 (0.039) | 0.714 (0.041) |
| | $M_{regular}$ | 0.508 (0.039) | 0.724 (0.042) | 0.726 (0.042) | 0.711 (0.044) | **0.761** (0.037) | 0.664 (0.041) | 0.701 (0.035) | 0.697 (0.042) | 0.467 (0.043) | 0.736 (0.037) |
| AUC (↑) | $M_{graphlets}$ | 0.508 (0.039) | 0.724 (0.042) | 0.726 (0.042) | **0.743** (0.039) | 0.725 (0.037) | 0.712 (0.039) | 0.658 (0.042) | 0.728 (0.039) | 0.494 (0.044) | 0.722 (0.040) |
| | $M_{reg+graphlets}$ | 0.508 (0.039) | 0.724 (0.042) | 0.726 (0.042) | **0.744** (0.039) | 0.677 (0.044) | 0.613 (0.042) | 0.700 (0.038) | 0.707 (0.041) | 0.493 (0.043) | 0.726 (0.038) |

Table 6: Cross-Task testbed results, obtained with different meta-graph features, for node classification-to-link prediction (top) and link prediction-to-node classification (bottom) settings. $M_{reg+graphlets}$ denotes a concatenation of $M_{regular}$ and $M_{graphlets}$ meta-graph features. Higher (↑) scores are better. The numbers in the parentheses denote one standard error. The **best** result is in bold, and the second best result is underlined.

(a) Node classification → Link prediction

| Perf. Metric | Meta-Feature | RandSel | GB-Perf | GB-Rank | ISAC | AS | S2 | ALORS | NCF | MetaOD | MetaGL |
|---|---|---|---|---|---|---|---|---|---|---|---|
| | $M_{compact}$ | 0.479 (0.016) | 0.652 (0.013) | **0.671** (0.013) | 0.646 (0.013) | 0.536 (0.016) | 0.626 (0.014) | 0.553 (0.014) | 0.670 (0.013) | 0.376 (0.016) | 0.601 (0.014) |
| | $M_{regular}$ | 0.479 (0.016) | 0.652 (0.013) | **0.671** (0.013) | 0.626 (0.014) | 0.520 (0.015) | 0.636 (0.014) | 0.594 (0.015) | 0.650 (0.013) | 0.465 (0.015) | 0.553 (0.014) |
| AUC (↑) | $M_{graphlets}$ | 0.479 (0.016) | 0.652 (0.013) | **0.671** (0.013) | 0.614 (0.014) | 0.510 (0.016) | 0.624 (0.014) | 0.526 (0.015) | 0.660 (0.014) | 0.409 (0.015) | 0.564 (0.014) |
| | $M_{reg+graphlets}$ | 0.479 (0.016) | 0.652 (0.013) | **0.671** (0.013) | 0.608 (0.015) | 0.525 (0.015) | 0.601 (0.014) | 0.582 (0.015) | 0.640 (0.013) | 0.414 (0.015) | 0.546 (0.015) |

(b) Link prediction → Node classification

| Perf. Metric | Meta-Feature | RandSel | GB-Perf | GB-Rank | ISAC | AS | S2 | ALORS | NCF | MetaOD | MetaGL |
|---|---|---|---|---|---|---|---|---|---|---|---|
| | $M_{compact}$ | 0.542 (0.025) | 0.595 (0.025) | 0.596 (0.025) | 0.557 (0.027) | 0.445 (0.027) | 0.490 (0.025) | 0.543 (0.025) | **0.617** (0.025) | 0.407 (0.024) | 0.554 (0.024) |
| | $M_{regular}$ | 0.542 (0.025) | 0.595 (0.025) | 0.596 (0.025) | 0.581 (0.026) | 0.517 (0.025) | 0.536 (0.024) | 0.606 (0.025) | **0.608** (0.025) | 0.477 (0.024) | 0.542 (0.026) |
| AUC (↑) | $M_{graphlets}$ | 0.542 (0.025) | 0.595 (0.025) | **0.596** (0.025) | 0.517 (0.029) | 0.477 (0.026) | 0.573 (0.025) | 0.505 (0.024) | 0.593 (0.025) | 0.460 (0.025) | 0.545 (0.025) |
| | $M_{reg+graphlets}$ | 0.542 (0.025) | 0.595 (0.025) | 0.596 (0.025) | 0.534 (0.028) | 0.494 (0.026) | 0.594 (0.025) | 0.518 (0.025) | **0.607** (0.026) | 0.448 (0.026) | 0.528 (0.027) |

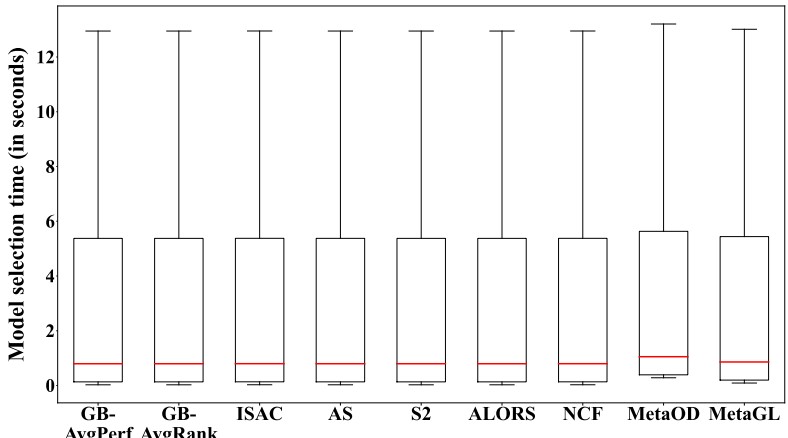

Figure 1: Time taken for selecting the best model (in seconds), which includes the time for generating meta-graph features for the new graph, and the time for model selection algorithms to infer the best model for the given graph based on them.

## A.3 Results on Time Cost

**Model Selection Runtime.** In Figure 1, we report the distribution of runtime (in seconds) for model selection, measured over the graphs in the benchmark. The runtime includes the time for generating meta-graph features for the new graph, and the time for a model selection method to infer the best model based on them. The median runtime (shown by the red line) is less than one second, and for a majority of the graphs, it takes at most five to six seconds. Note that the distributions are mostly the same for different methods, as the time for different model selection algorithms to infer the best model is very short (close to zero second), and similar to each other. We measured the time for meta feature generation via sequential processing for simplicity, while these features can be processed in parallel as they are independent of each other.

**Training Runtime.** Table 7 reports the time (in seconds) taken for training model selection algorithms until convergence on the Fully-Observed testbed for the link prediction task. Note that Global Best algorithms and AS are excluded as they do not require model training. ISAC takes the least amount of time as it only performs clustering without model parameter updates during the training phase. Algorithms that rely on neural networks require more training time. Yet a majority of them can still be trained quickly, in just a few seconds to a few minutes. MetaOD takes the largest amount of time with its current optimization framework.

Table 7: Time (in seconds) taken for training model selection algorithms on the Fully-Observed testbed for the link prediction task. For each algorithm, we show the training runtime averaged over the splits in the testbed, and the standard deviation in the parentheses.

| ISAC | S2 | ALORS | NCF | MetaOD | MetaGL |
|---|---|---|---|---|---|
| 0.0330 | 0.6162 | 0.7530 | 9.2510 | 8136.1713 | 172.0742 |
| (0.0279) | (0.6473) | (0.0897) | (2.3227) | (89.6785) | (50.9201) |

## A.4 Testbed Results With Standard Error

In the main text, we provide the average model selection performances in four testbeds, *i.e.*, Fully-Observed, Sparse, Out-Of-Domain, and Small-To-Large testbeds, but their standard errors could not be shown due to space constraint. In this subsection, we present the standard error along with the average performance in those four testbeds (Tables 8 to 11). Please refer to the main text for the discussion of the results of these testbeds. Cross-Task testbed results are discussed in Appendix A.1

Table 8: Fully-Observed testbed results for link prediction (top) and node classification (bottom) tasks. Higher (↑) scores are better. The numbers in the parentheses denote one standard error. The **best** result is in bold, and the second best result is underlined.

(a) Link prediction

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

# B  Model Selection Algorithms

Here we provide details of the model selection algorithms included in GLEMOS.

**Random Selection** assigns random scores to the models, and thus chooses the best model purely randomly without considering prior model performances and meta-graph features.

**Global Best (GB)-AvgPerf** [27] computes the average performance of each model over all observed graphs, and selects the one with the largest average performance. Thus this algorithm is independent of meta-graph features.

**Global Best (GB)-AvgRank** [18] computes the rank (in percentile) of all models for each graph, where a higher performance is assigned a larger rank percentile, and selects the model that has the largest average rank percentile over observed graphs. Thus this algorithm is also independent of meta-graph features.

**ISAC** [12] clusters observed graphs into $k$ groups in the space of meta-graph features, and when given a new graph, finds the cluster nearest to the given graph and selects the model that obtained the highest average performance over the observed graphs in that cluster. $k$ is set to 5 in our experiments.

**ARGOSMART (AS)** [17] finds the observed graph, which is the most similar to the test graph in terms of meta-graph features, and selects the model that had the best performance on that graph.

**Supervised Surrogates (S2)** [25] optimizes a surrogate model (*i.e.*, a regressor) to transform meta-graph features into model performances. We use a two-layer feedforward neural network with ReLU nonlinearity as the surrogate model.

**ALORS** [16] learns latent factors of the observed graphs and models by performing low-rank nonnegative matrix factorization on the performance matrix, and then optimizes a non-linear regressor to map meta-graph features into latent graph factors. Given a new graph, it predicts the new graph's latent factor, and estimates the model performances on the new graph to be the dot product between the estimated graph factor and the learned model factor.

**NCF** [10] adapts ALORS by replacing the dot product operation used in ALORS with a more flexible neural network model, which predicts model performances by jointly employing the linearity of matrix factorization and nonlinearity of deep neural networks.

**MetaOD** [27] improves ALORS for the model selection task, *e.g.*, by employing a rank-based meta-learning objective instead of the usual reconstruction objective. As the first method for unsupervised outlier model selection, it also presents specialized meta-features to capture the outlying characteristics of non-graph data.

**MetaGL** [18] extends MetaOD for GL model selection by designing meta-graph features to capture the structural characteristics of a graph, employing the top-1 probability as a meta-training objective, and modeling the relations between graphs and models in the form of a heterogeneous graph and learning latent factors by applying graph neural networks on it.

## C   Meta-Graph Features

### C.1   Time Complexity Analysis

Let $G = (V, E)$ be a graph, where $V$ and $E$ denote the set of nodes and edges in graph $G$, respectively. Overall, the time complexity of generating meta-graph features for graph $G$ is

$$\mathcal{O}(k|E|\Delta) \tag{1}$$

where $k$ is the number of feature extractors in the extractor set $|\Psi|$, and $\Delta$ denotes a small constant.

Each feature extractor in $\Psi$ extracts a specific graph structural feature, such as network motifs and PageRank scores, as well as other graph statistics such as the density of a graph. We estimate the frequency of all network motifs with $\{2, 3, 4\}$-nodes, which can be done in $\mathcal{O}(|E|\Delta)$ time, where $\Delta$ is a small constant representing the maximum sampled degree, which can be set by the user. For further details, please refer to [1]. Other structural features, such as PageRank, take $\mathcal{O}(|E|)$ time, and graph statistics, such as the number of nodes and edges and the graph density, can be obtained even more efficiently. Assuming $k$ such feature extractors, overall it takes $\mathcal{O}(k|E|\Delta)$ time.

Note that GLEMOS aims to provide a representative and diverse set of graph features, which have been proven effective in previous studies. At the same time, our framework is flexible, and can support any set of meta-graph features. Thus, depending on the task, more computationally expensive, yet informative new features might be additionally used, or the set of features may be limited to only those that can be efficiently computed in time strictly linear in the number of edges, *i.e.*, $\mathcal{O}(|E|)$, in which case the $\Delta$ term would be dropped, and we have $\mathcal{O}(k|E|)$. Further, note that feature extractors are independent of each other, and can be computed in parallel.

### C.2   Statistical Functions

Table 12 lists the statistical functions $\Sigma$, which derives a set of meta-graph features that summarize the statistical distribution of graph invariants, such as the node degree distribution, or $k$-core numbers. Such vectors have different sizes for different graphs as their size is determined by the number of nodes or edges of the graph. The statistical functions $\Sigma$ in Table 12 transforms those vectors of varying length into fixed-size meta-feature vectors.

Table 12: Statistical functions $\Sigma$ to derive a set of meta-graph features from a graph invariant, *e.g.*, degree distribution or $k$-core numbers. $\mathbf{x}$ denotes a vector of arbitrary graph invariants for some graph $G_i = (V_i, E_i)$, such as node degree vector, and PageRank vector (*i.e.*, PageRank scores of each node in $G_i$). $\pi(\mathbf{x})$ denotes the sorted vector of $\mathbf{x}$.

| Function | Equation |
| --- | --- |
| Min, Max | $\min(\mathbf{x})$, $\max(\mathbf{x})$ |
| Median | $\mathrm{med}(\mathbf{x})$ |
| Geometric Mean | $\|\mathbf{x}\|^{-1} \prod_i x_i$ |
| Harmonic Mean | $\|\mathbf{x}\| / \sum_i \frac{1}{x_i}$ |
| Mean, Stdev, Variance | $\mu_{\mathbf{x}}, \sigma_{\mathbf{x}}, \sigma_{\mathbf{x}}^2$ |
| Skewness | $\mathbb{E}(\mathbf{x}-\mu_{\mathbf{x}})^3 / \sigma_{\mathbf{x}}^3$ |
| Pearson Kurtosis | $\mathrm{Kurt}[\mathbf{x}] = \mathbb{E}(\mathbf{x}-\mu_{\mathbf{x}})^4 / \sigma_{\mathbf{x}}^4$ (biased/unbiased) |
| Fisher Kurtosis | $\mathrm{Kurt}[\mathbf{x}] - 3.0$ (biased/unbiased) |
| Quartile Dispersion Coeff. | $\frac{Q_3 - Q_1}{Q_3 + Q_1}$ |
| Median Absolute Deviation | $\mathrm{med}(\|\mathbf{x} - \mathrm{med}(\mathbf{x})\|)$ |
| Avg. Absolute Deviation | $\frac{1}{\|\mathbf{x}\|} \mathbf{e}^T \|\mathbf{x} - \mu_{\mathbf{x}}\|$ |
| Coeff. of Variation | $\sigma_{\mathbf{x}}/\mu_{\mathbf{x}}$ |
| Efficiency Ratio | $\sigma_{\mathbf{x}}^2/\mu_{\mathbf{x}}^2$ |
| Variance-to-Mean Ratio | $\sigma_{\mathbf{x}}^2/\mu_{\mathbf{x}}$ |
| Signal-to-Noise Ratio (SNR) | $\mu_{\mathbf{x}}^2/\sigma_{\mathbf{x}}^2$ |
| Entropy | $H(\mathbf{x}) = -\sum_i x_i \log x_i$ |
| Norm. Entropy | $H(\mathbf{x})/\log_2 \|\mathbf{x}\|$ |
| Gini Coefficient | $\sum_{i=1}^{\|\mathbf{x}\|} (2i - \|\mathbf{x}\| - 1)\pi(\mathbf{x})_i / n \sum_{i=1}^{\|\mathbf{x}\|} \pi(\mathbf{x})_i$ |
| $Q_1, Q_3$ | median of the $\|\mathbf{x}\|/2$ smallest (largest) values |
| IQR | $Q_3 - Q_1$ |
| Outlier LB $\alpha \in \{1.5, 3\}$ | $Q_1 - \alpha IQR$ |
| Outlier UB $\alpha \in \{1.5, 3\}$ | $Q_3 + \alpha IQR$ |
| Outliers Count $\alpha \in \{1.5, 3\}$, $(\beta, \gamma) \in \{(1,0), (0,1), (1,1)\}$ | $\beta \cdot \sum_i \mathbb{I}(x_i < Q_1 - \alpha IQR) + \gamma \cdot \sum_i \mathbb{I}(x_i > Q_3 + \alpha IQR)$ |
| Outliers Frac. $\alpha \in \{1.5, 3\}$, $(\beta, \gamma) \in \{(1,0), (0,1), (1,1)\}$ | $(\beta \cdot \sum_i \mathbb{I}(x_i < Q_1 - \alpha IQR) + \gamma \cdot \sum_i \mathbb{I}(x_i > Q_3 + \alpha IQR))/\|\mathbf{x}\|$ |
| ($\alpha$-std) Outliers Count $\alpha \in \{1, 2, 3\}$, $(\beta, \gamma) \in \{(1,0), (0,1), (1,1)\}$ | $\beta \cdot \sum_i \mathbb{I}(x_i < \mu_{\mathbf{x}} - \alpha\sigma_{\mathbf{x}}) + \gamma \cdot \sum_i \mathbb{I}(x_i > \mu_{\mathbf{x}} + \alpha\sigma_{\mathbf{x}})$ |
| ($\alpha$-std) Outliers Frac. $\alpha \in \{1, 2, 3\}$, $(\beta, \gamma) \in \{(1,0), (0,1), (1,1)\}$ | $(\beta \cdot \sum_i \mathbb{I}(x_i < \mu_{\mathbf{x}} - \alpha\sigma_{\mathbf{x}}) + \gamma \cdot \sum_i \mathbb{I}(x_i > \mu_{\mathbf{x}} + \alpha\sigma_{\mathbf{x}}))/\|\mathbf{x}\|$ |
| Mode | modal (most common) value in $\mathbf{x}$ |
| Mode Count | count for the modal value in $\mathbf{x}$ |
| Mode Frac. | mode count of $\mathbf{x}$ / $\|\mathbf{x}\|$ |

## D  Experimental Settings and Details

**Hardware.** Experiments were performed on a Linux server on AWS, running Ubuntu 20.04.5 LTS with Intel Xeon Platinum 8275CL CPUs @ 3.00GHz, 1.1TB RAM, and NVIDIA A100 SXM4 GPUs with 40GB memory.

**Performance Evaluation.** To evaluate the performance of optimizable GL methods, such as GCN [13] and GraphSAGE [9], we trained these methods for up to 300 epochs, using Adam optimizer with a learning rate of 0.001, and applying a validation-based early stopping with a patience of 30 epochs. A few graphs have multiple labels for each node. For those graphs, we found that using a larger learning rate and patience leads to a better performance. So for the multi-label node classification datasets, we used a learning rate of 0.01 and a patience of 60 epochs. As an early stopping criterion, we used ROC AUC for link prediction, and used accuracy for node classification (or weighted average precision for multi-label node classification). Not all graphs come with input node features. For those graphs without input node features, we used randomly initialized embeddings of size 32 as input node embeddings, and let those embeddings optimized during model training.

**Meta-Graph Features.** We present three sets of meta-graph features in the main text, *i.e.*, $M_{regular}$, $M_{graphlets}$, and $M_{compact}$. The experimental results reported in the main text were obtained using $M_{regular}$. We also report results obtained with three different sets of meta-graph features, *i.e.*, $M_{graphlets}$, $M_{compact}$, and $M_{reg+graphlets}$, in Appendix A.2.

**Graph Learning (GL) Methods.** In the evaluation using the proposed testbeds, model selection algorithms aim to predict the best model from the set of differently configured GL models, that is, GCN [13], GraphSAGE [9], GAT [21], GIN [24], EGC [20], SGC [23], ChebNet [5], PNA [4], DGI [22], spectral embedding [15], GraRep [2], node2vec [7], label propagation [29], Jaccard's Coeff. [14], Resource Alloc. [28], Adamic/Adar [14], and SEAL [26]. Their hyperparameter settings are provided in Table 2 in the main text. We implemented spectral embedding [15] and GraRep [2], using NumPy[2] and SciPy[3]. We used GRAPE [3] for the implementation of node2vec [7]. We used NetworkX [8] for the implementation of classical link prediction methods, *i.e.*, Jaccard's Coeff. [14], Resource Alloc. [28], and Adamic/Adar [14]. We used PyTorch Geometric (PyG) [6] for the implementation of other GL methods, *e.g.*, GCN [13] and GraphSAGE [9]. Spectral embedding and GraRep use SciPy's functionalities to find eigenvalue/eigenvectors and perform singular value decomposition, respectively; for these methods, we used the default parameter values specified by SciPy, except for the parameters that we vary in creating the model set $\mathcal{M}$. Similarly, for methods supported by PyG, we used their default parameter settings in the corresponding package, while varying a few important parameters to create the model set $\mathcal{M}$.

**Model Selection Algorithms.** In experiments, we evaluate ten model selection algorithms, that is, Random Selection, Global Best-AvgPerf [27], Global Best-AvgRank [18], ISAC [12], ARGOSMART (AS) [17], Supervised Surrogates (S2) [25], ALORS [16], NCF [10], MetaOD [27], and MetaGL [18]. For MetaGL [18] and MetaOD [27], we used the authors' implementation. We adapted MetaOD's implementation so that it can work with a sparse performance matrix. We implemented other model selection algorithms included in GLEMOS in python using open source libraries such as NumPy[2] and DGL[4]. Global Best methods perform a global averaging of the performance matrix. Given sparse performance matrices, they average over observed entries alone and ignore missing entries. ISAC [12] applies $k$-means algorithm to meta-graph features to cluster observed graphs into 5 groups. AS [17] uses cosine similarity scoring to find the 1-NN observed graph. S2 [25] uses Adam optimizer to train an MLP regressor with two hidden layers, which is optimized to transform meta-graph features into model performances. ALORS [16] learns latent embeddings by using nonnegative matrix factorization, and uses an MLP regressor with two hidden layers to transform meta-graph features into latent graph factors. NCF [10] produces latent graph and model embedding by using an MLP regressor with two hidden layers, which is optimized via Adam optimizer with a learning rate of 0.01 and a weight decay of 0.0001. MetaOD [27] uses the default parameter settings given by the original implementation, *e.g.*, a random forest regressor with 100 estimators and a max depth of 10. MetaGL [18] uses heterogeneous graph transformer (HGT) [11] as a graph encoder, which contains 2 layers and 4 attention heads per layer. In its G-M network, nodes are connected to their top-30 most similar nodes. The MetaGL model is optimized with Adam optimizer with a learning rate of 0.00075 and a weight decay of 0.0001. For optimizable algorithms discussed above, which involve learning low-dimensional embeddings, we consistently set the embedding size to 32.

# E   Usage and Extensibility

We describe how GLEMOS can be used and extended with graphs, models, and performance records.

## E.1   Graphs

To support graph data from multiple data sources, graphs in GLEMOS are represented by the `graphs.graphset.Graph` class, which is a wrapper class that holds the graph data from different sources, and provides auxiliary methods serving as a common interface to different types of graph data.

The set of graphs in GLEMOS is represented by the `graphs.graphset.GraphSet` class, which provides the functionality to load graphs for a certain graph learning task like node classification, as well as graphs from specific data sources, such as the Network Repository (NetRepo) [19] and PyTorch Geometric (PyG) [6], as shown below.

---

[2] https://numpy.org/
[3] https://scipy.org/
[4] http://dgl.ai/

```
1  from graphs.graphset import GraphSet
2  graph_set = GraphSet(['netrepo', 'pyg'])
3  node_classification_graphs = graph_set.node_classification_graphs()
```

**Adding New Graphs From Existing Sources.** We currently provide code to incorporate graphs from the two popular graph repositories, *i.e.*, NetRepo and PyG. Adding new graphs from these repositories can be done easily by instantiating a `graphs.graphset.Graph` object for the new graph, and adding it to the list returned by the `load_graphs` functions (e.g,. `graphs.pyg_graphs.load_graphs` )

**Adding New Graphs From New Sources** (*e.g.*, a new graph repository or your own graph data) can be done by (1) adding a script for the new data source in the `graphs` package, which will parse the raw graph data, and construct a `graphs.graphset.Graph` object for the new graph, and (2) registering the new data source in the `graphs.graphset.GraphSet` class.

## E.2   Models

The set of models included in GLEMOS is defined by the `models.modelset` package. Each graph learning method and its hyperparameter settings to be searched over are defined by a separate class that inherits from the `models.modelset.ModelSettings` class. For example, the following code defines the GAT model set.

```
1  class GATModelSettings(ModelSettings):
2    def __init__(self):
3      super().__init__()
4      self.variable_hyperparams = ModelSettings.alpha_ordered_dict({
5        'hidden_channels': [16, 64],
6        'num_layers': [1, 2, 3],
7        'dropout': [0.0, 0.5],
8        'heads': [1, 4],
9        'concat': [True, False],
10     })
11
12   @classmethod
13   def load_model(cls, in_channels, out_channels, **params):
14     return GAT(in_channels=in_channels, out_channels=out_channels, **
       params)
```

In the above code snippet, a GAT model instance, instantiated with particular hyperparameter settings, is obtained via the `models.modelset.ModelSettings.load_model` method.

**Adding New Hyperparameter Settings to Existing Models** can be done by simply adding additional hyperparameter settings to the corresponding `models.modelset.ModelSettings` class.

**Adding New Models** can be done by (1) creating a new `ModelSettings` class for the new graph learning model, (2) specifying the set of hyperparameter settings to be searched over, and (3) completing the `load_model` method, such that new instantiated model object is to be returned, as in the above code.

## E.3   Performance Records

**Data Splits.** For consistent comparisons among different models, GLEMOS provides the data splits used in the evaluation. The `graphs.datasplit.DataSplit` class provides functionalities to generate and load data splits (*e.g.*, node splits for node classification, and edge splits for link prediction), and the generated data splits are included in GLEMOS. Then these previously generated data splits are used when evaluating new GL models on the existing graphs in the benchmark.

**Model Training and Evaluation.** Given the instantiated GL model and the graph data, model training and evaluation is taken care of by the `models.trainer.Trainer` class. The same code can be used to train and evaluate new GL models on either existing or new graphs,

**Parallel Processing.** To perform the training and evaluation of multiple <model, graph> pairs, the aforementioned `models.trainer.Trainer` class is repeatedly executed by the `performances.taskrunner.Runner` class. The `Runner` class is designed to support parallel pro-

cessing, such that multiple processes pick up an unevaluated <model, graph> pair, and perform model training and evaluation in parallel.

# F   Additional Details of the Dataset

## F.1   Data Overview

GLEMOS provides the following sets of data.

(a) **Performance records**: performance of graph learning (GL) models on various graphs, measured in multiple metrics for the following GL tasks.
- Node classification performances
- Link prediction performances

(b) **Graph data splits** used for evaluating GL models.
- node splits for node classification
- edge splits for link prediction

(c) **Testbed data splits** (*i.e.*, splitting over the performance matrix and the meta-graph features) to evaluate model selection algorithms.

(d) **Meta-graph features**: we provide the following sets of meta-graph features.
- $M_{regular}$
- $M_{graphlets}$
- $M_{compact}$
- $M_{reg+graphlets}$

Performance records (a), splitting of graph data (b) and testbed data (c), and meta-graph features (d) were generated by GLEMOS by processing the graph data (Appendix F.2). For details of these four types of data, please refer to the main text. We give further description of the graph data used in the benchmark in the next subsection.

**No Personal or Offensive Contents.**   Note that GLEMOS does not include personal data (*e.g.*, personally identifiable information) or offensive contents in all the data described above ((a) to (d)).

## F.2   Graph Data

**Graph Data Sources.**   We collect graph data from two widely used public graph repositories, *i.e.*, Network Repository (NetRepo) [19] and PyTorch Geometric (PyG) [6]. In Table 14, we provide the list of graphs in GLEMOS, and specify the data source for each graph.

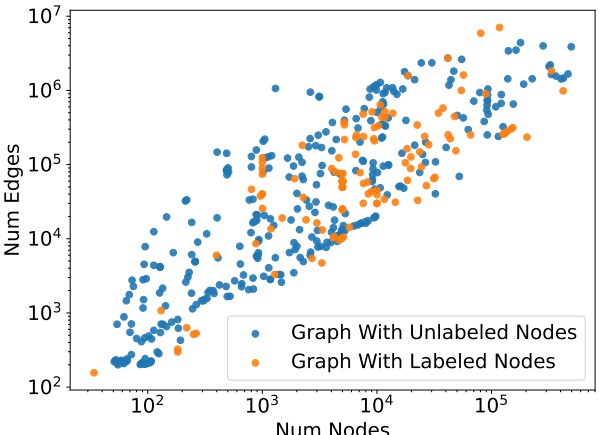

Figure 2: Distribution of the graphs in terms of the number of nodes (x-axis) and edges (y-axis). Each dot corresponds to a graph, and is colored depending on whether the nodes in the graph have class labels or not.

Table 13: Distribution of the graphs in GLEMOS per graph domain. In total, GLEMOS currently covers 457 graphs drawn from 37 domains.

| Graph Domain | Count | Graph Domain | Count | Graph Domain | Count |
|---|---|---|---|---|---|
| Social Networks | 53 | Synthetic-ER | 15 | Synthetic-SBM | 6 |
| Chemical | 40 | Temporal Reachability | 14 | Road | 5 |
| Protein | 26 | Synthetic-RandPart | 12 | Recommendation | 5 |
| Retweet | 26 | Interaction | 10 | Computer Vision | 4 |
| Biological Networks | 24 | Proximity | 10 | Flight | 3 |
| Web Graphs | 22 | Brain Networks | 9 | Misc | 3 |
| Economic Networks | 17 | Power Networks | 8 | Co-Purchase | 3 |
| Synthetic-BA | 17 | Citation | 8 | Scientific Computing | 3 |
| Friendship | 16 | Email | 7 | Infrastructure | 2 |
| Synthetic-CL | 16 | Technology | 7 | Coauthor | 2 |
| Collaboration | 15 | Ecology | 6 | Phone Call Networks | 1 |
| Synthetic-KPGM | 15 | Wikipedia | 6 | | |
| Synthetic-Others | 15 | Knowledgebase | 6 | | |
| | | | | **Total Number of Graphs** | **457** |

**Graph Data Overview.** Figure 2 shows the distribution of the graph in GLEMOS in terms of the number of nodes (x-axis) and the number of edge (y-axis), where each dot corresponds to a graph, and is colored depending on whether nodes are labeled or not. Table 13 shows the distribution of the graphs per graph domain. In total, 457 graphs drawn from 37 domains are currently used for model performance evaluation.

**List of Graphs.** Table 14 lists the graphs in GLEMOS, and provides information on the graph size, the number of node classes, data source, and graph domain.

Table 14: List of the graphs.

| Id | Graph | # Nodes | # Edges | # Node Classes | Data Source | Domain |
|---|---|---|---|---|---|---|
| 1 | PLC-60-30-L2 | 117,572 | 7,045,181 | 2 | NetRepo | Synthetic-Others |
| 2 | soc-Flickr-ASU | 80,513 | 5,899,882 | 195 | NetRepo | Social Networks |
| 3 | sc-shipsec5 | 179,104 | 4,400,152 | N/A | NetRepo | Scientific Computing |
| 4 | web-Stanford | 281,903 | 3,985,272 | N/A | NetRepo | Web Graphs |
| 5 | soc-youtube | 495,957 | 3,873,496 | N/A | NetRepo | Social Networks |
| 6 | web-arabic-2005 | 163,598 | 3,494,538 | N/A | NetRepo | Web Graphs |
| 7 | sc-shipsec1 | 140,385 | 3,415,518 | N/A | NetRepo | Scientific Computing |
| 8 | LINKX-penn94 | 41,554 | 2,724,458 | 2 | PyG | Social Networks |
| 9 | socfb-Penn94 | 41,536 | 2,724,440 | N/A | NetRepo | Social Networks |
| 10 | sc-nasasrb | 54,870 | 2,622,454 | N/A | NetRepo | Scientific Computing |
| 11 | socfb-Michigan23 | 30,147 | 2,353,032 | N/A | NetRepo | Social Networks |
| 12 | socfb-UGA50 | 24,389 | 2,348,114 | N/A | NetRepo | Social Networks |
| 13 | web-NotreDame | 325,729 | 2,207,671 | N/A | NetRepo | Web Graphs |
| 14 | ca-dblp-2012 | 317,080 | 2,099,732 | N/A | NetRepo | Collaboration |
| 15 | Entities-BGS | 333,845 | 1,832,398 | 2 | PyG | Knowledgebase |
| 16 | KPGM-log16-16-trial1 | 46,872 | 1,818,168 | N/A | NetRepo | Synthetic-KPGM |
| 17 | socfb-Oklahoma97 | 17,425 | 1,785,056 | N/A | NetRepo | Social Networks |
| 18 | soc-twitter-follows-mun | 465,017 | 1,667,082 | N/A | NetRepo | Social Networks |
| 19 | ca-MathSciNet | 332,689 | 1,641,288 | N/A | NetRepo | Collaboration |
| 20 | AttributedGraph-PPI | 56,944 | 1,612,348 | 121 | PyG | Protein |
| 21 | socfb-Cornell5 | 18,660 | 1,581,554 | N/A | NetRepo | Social Networks |
| 22 | LINKX-cornell5 | 18,660 | 1,581,554 | 2 | PyG | Friendship |
| 23 | ia-dbpedia-team-bi | 365,492 | 1,560,266 | N/A | NetRepo | Interaction |
| 24 | rt-higgs | 425,008 | 1,465,617 | N/A | NetRepo | Retweet |
| 25 | ca-dblp-2010 | 226,413 | 1,432,920 | N/A | NetRepo | Collaboration |
| 26 | ia-wiki-trust-dir | 138,592 | 1,432,057 | N/A | NetRepo | Interaction |
| 27 | soc-twitter-follows | 404,719 | 1,426,638 | N/A | NetRepo | Social Networks |
| 28 | socfb-Virginia63 | 21,325 | 1,396,356 | N/A | NetRepo | Social Networks |
| 29 | KPGM-log16-12-trial1 | 44,241 | 1,395,310 | N/A | NetRepo | Synthetic-KPGM |
| 30 | BA-2_3_60 | 10,708 | 1,281,300 | N/A | NetRepo | Synthetic-BA |
| 31 | tech-RL-caida | 190,914 | 1,215,220 | N/A | NetRepo | Technology |
| 32 | BA-2_21_60 | 9,993 | 1,195,500 | N/A | NetRepo | Synthetic-BA |
| 33 | KPGM-log16-10-trial1 | 42,551 | 1,177,546 | N/A | NetRepo | Synthetic-KPGM |
| 34 | BA-2_20_60 | 9,691 | 1,159,260 | N/A | NetRepo | Synthetic-BA |
| 35 | BA-2_15_60 | 9,340 | 1,117,140 | N/A | NetRepo | Synthetic-BA |

*Continued on the next page*

| Id | Graph | # Nodes | # Edges | # Node Classes | Data Source | Domain |
|---|---|---|---|---|---|---|
| 36 | socfb-Syracuse56 | 13,653 | 1,087,964 | N/A | NetRepo | Social Networks |
| 37 | socfb-NotreDame57 | 12,155 | 1,082,678 | N/A | NetRepo | Social Networks |
| 38 | scc_fb-messages | 1,303 | 1,063,786 | N/A | NetRepo | Temporal Reachability |
| 39 | socfb-UC33 | 16,808 | 1,044,294 | N/A | NetRepo | Social Networks |
| 40 | CL-100000-1d7-trial3 | 92,967 | 1,043,480 | N/A | NetRepo | Synthetic-CL |
| 41 | BA-2_9_60 | 8,717 | 1,042,380 | N/A | NetRepo | Synthetic-BA |
| 42 | socfb-Duke14 | 9,885 | 1,012,874 | N/A | NetRepo | Social Networks |
| 43 | GemsecDeezer-HR | 54,573 | 996,404 | 84 | PyG | Friendship |
| 44 | LINKX-genius | 421,961 | 984,979 | 2 | PyG | Social Networks |
| 45 | socfb-JMU79 | 14,070 | 971,128 | N/A | NetRepo | Social Networks |
| 46 | scc_twitter-copen | 2,623 | 947,228 | N/A | NetRepo | Temporal Reachability |
| 47 | BA-2_23_40 | 11,770 | 939,960 | N/A | NetRepo | Synthetic-BA |
| 48 | soc-slashdot-zoo | 79,120 | 935,738 | N/A | NetRepo | Social Networks |
| 49 | BA-2_11_40 | 11,337 | 905,320 | N/A | NetRepo | Synthetic-BA |
| 50 | Flickr | 89,250 | 899,756 | 7 | PyG | Computer Vision |
| 51 | socfb-UCSD34 | 14,948 | 886,442 | N/A | NetRepo | Social Networks |
| 52 | rec-github | 121,709 | 879,770 | N/A | NetRepo | Recommendation |
| 53 | CL-100000-1d8-trial3 | 92,402 | 871,580 | N/A | NetRepo | Synthetic-CL |
| 54 | econ-psmigr3 | 3,140 | 824,702 | N/A | NetRepo | Economic Networks |
| 55 | econ-psmigr1 | 3,140 | 824,702 | N/A | NetRepo | Economic Networks |
| 56 | econ-psmigr2 | 3,140 | 821,562 | N/A | NetRepo | Economic Networks |
| 57 | socfb-Yale4 | 8,578 | 810,900 | N/A | NetRepo | Social Networks |
| 58 | CL-100000-1d9-trial3 | 91,889 | 742,469 | N/A | NetRepo | Synthetic-CL |
| 59 | ia-wiki-Talk | 92,117 | 721,534 | N/A | NetRepo | Interaction |
| 60 | soc-slashdot | 70,068 | 717,294 | N/A | NetRepo | Social Networks |
| 61 | socfb-Cal65 | 11,247 | 702,716 | N/A | NetRepo | Social Networks |
| 62 | web-sk-2005 | 121,422 | 668,838 | N/A | NetRepo | Web Graphs |
| 63 | soc-douban | 154,908 | 654,324 | N/A | NetRepo | Social Networks |
| 64 | ER-2_2_50 | 11,429 | 651,760 | N/A | NetRepo | Synthetic-ER |
| 65 | BA-2_24_60-L2 | 10,693 | 639,750 | 2 | NetRepo | Synthetic-BA |
| 66 | CL-100000-2d0-trial1 | 91,471 | 631,153 | N/A | NetRepo | Synthetic-CL |
| 67 | ER-3_19_50 | 108,999 | 600,554 | N/A | NetRepo | Synthetic-ER |
| 68 | ER-2_21_50 | 10,804 | 584,286 | N/A | NetRepo | Synthetic-ER |
| 69 | GitHub | 37,700 | 578,006 | 2 | PyG | Friendship |
| 70 | socfb-Tulane29 | 7,752 | 567,836 | N/A | NetRepo | Social Networks |
| 71 | socfb-Wake73 | 5,372 | 558,382 | N/A | NetRepo | Social Networks |
| 72 | CL-100000-2d1-trial2 | 90,880 | 543,035 | N/A | NetRepo | Synthetic-CL |
| 73 | HeterophilousGraph-Tolokers | 11,758 | 519,000 | 2 | PyG | Collaboration |
| 74 | web-spam-detection | 9,072 | 514,700 | 3 | NetRepo | Web Graphs |
| 75 | ER-2_8_50 | 10,070 | 506,096 | N/A | NetRepo | Synthetic-ER |
| 76 | Coauthor-Physics | 34,493 | 495,924 | 5 | PyG | Coauthor |
| 77 | Amazon-Computers | 13,752 | 491,722 | 10 | PyG | Co-Purchase |
| 78 | AttributedGraph-Flickr | 7,575 | 479,476 | 9 | PyG | Social Networks |
| 79 | ia-wikiquote-user-edits | 93,445 | 476,865 | N/A | NetRepo | Interaction |
| 80 | rec-yelp-user-business | 50,395 | 459,208 | N/A | NetRepo | Recommendation |
| 81 | GemsecDeezer-HU | 47,538 | 445,774 | 84 | PyG | Friendship |
| 82 | PLC-40-30-L5 | 11,025 | 437,979 | 5 | NetRepo | Synthetic-Others |
| 83 | WikiCS | 11,701 | 431,726 | 10 | PyG | Wikipedia |
| 84 | soc-brightkite | 56,739 | 425,890 | N/A | NetRepo | Social Networks |
| 85 | KPGM-log14-16-trial3 | 12,545 | 425,872 | N/A | NetRepo | Synthetic-KPGM |
| 86 | socfb-UChicago30 | 6,591 | 416,206 | N/A | NetRepo | Social Networks |
| 87 | web-wiki-squirrel | 5,201 | 396,846 | N/A | NetRepo | Web Graphs |
| 88 | ca-AstroPh | 17,903 | 393,944 | N/A | NetRepo | Collaboration |
| 89 | ER-2_14_50 | 8,851 | 390,774 | N/A | NetRepo | Synthetic-ER |
| 90 | ER-3_18_50 | 86,337 | 381,446 | N/A | NetRepo | Synthetic-ER |
| 91 | LINKX-johnshopkins55 | 5,180 | 373,172 | 2 | PyG | Social Networks |
| 92 | socfb-Rice | 4,087 | 369,656 | N/A | NetRepo | Social Networks |
| 93 | scc_infect-dublin | 10,972 | 351,146 | N/A | NetRepo | Temporal Reachability |
| 94 | AttributedGraph-BlogCatalog | 5,196 | 343,486 | 6 | PyG | Social Networks |
| 95 | FacebookPagePage | 22,470 | 342,004 | 4 | PyG | Web Graphs |
| 96 | web-wiki-crocodile | 11,631 | 341,691 | N/A | NetRepo | Web Graphs |
| 97 | soc-BlogCatalog-ASU | 10,312 | 333,983 | 39 | NetRepo | Social Networks |
| 98 | KPGM-log14-12-trial1 | 11,893 | 330,072 | N/A | NetRepo | Synthetic-KPGM |
| 99 | road-usroads-48 | 126,146 | 323,900 | N/A | NetRepo | Road |
| 100 | Twitch-DE | 9,498 | 315,774 | 2 | PyG | Friendship |
| 101 | Tox21-p53 | 153,563 | 314,046 | 47 | NetRepo | Chemical |
| 102 | socfb-UC64 | 6,833 | 310,664 | N/A | NetRepo | Social Networks |
| 103 | Tox21-AHR | 147,772 | 302,188 | 49 | NetRepo | Chemical |
| 104 | tech-p2p-gnutella | 62,561 | 295,756 | N/A | NetRepo | Technology |
| 105 | Tox21-HSE | 136,239 | 277,682 | 47 | NetRepo | Chemical |
| 106 | socfb-Wesleyan43 | 3,593 | 276,070 | N/A | NetRepo | Social Networks |
| 107 | Mutagenicity | 131,488 | 266,894 | 14 | NetRepo | Chemical |
| 108 | Tox21-MMP | 127,998 | 260,962 | 47 | NetRepo | Chemical |
| 109 | Tox21-aromatase | 126,483 | 257,092 | 46 | NetRepo | Chemical |
| 110 | GemsecDeezer-RO | 41,773 | 251,652 | 84 | PyG | Friendship |

| Id | Graph | # Nodes | # Edges | # Node Classes | Data Source | Domain |
|---|---|---|---|---|---|---|
| 111 | NELL | 65,755 | 251,550 | 186 | PyG | Knowledgebase |
| 112 | rec-amazon | 91,813 | 251,408 | N/A | NetRepo | Recommendation |
| 113 | fb-CMU-Carnegie49 | 6,637 | 249,967 | 3 | NetRepo | Social Networks |
| 114 | socfb-Middlebury45 | 3,075 | 249,220 | N/A | NetRepo | Social Networks |
| 115 | DBLP | 26,128 | 239,566 | 4 | PyG | Knowledgebase |
| 116 | road-luxembourg-osm | 114,599 | 239,332 | N/A | NetRepo | Road |
| 117 | Amazon-Photo | 7,650 | 238,162 | 8 | PyG | Co-Purchase |
| 118 | ca-HepPh | 11,204 | 235,238 | N/A | NetRepo | Collaboration |
| 119 | EllipticBitcoin | 203,769 | 234,355 | 3 | PyG | Economic Networks |
| 120 | Twitch-FR | 6,551 | 231,883 | 2 | PyG | Friendship |
| 121 | KPGM-log14-8-trial1 | 10,978 | 227,990 | N/A | NetRepo | Synthetic-KPGM |
| 122 | socfb-Trinity100 | 2,613 | 223,992 | N/A | NetRepo | Social Networks |
| 123 | MSRC-21 | 43,644 | 223,312 | 22 | NetRepo | Computer Vision |
| 124 | DHFR-MD | 9,380 | 222,452 | 7 | NetRepo | Chemical |
| 125 | ER-1_5_20 | 1,050 | 220,058 | N/A | NetRepo | Synthetic-ER |
| 126 | bio-HS-CX | 4,413 | 217,636 | N/A | NetRepo | Biological Networks |
| 127 | WikipediaNetwork-Squirrel | 5,201 | 217,073 | 5 | PyG | Wikipedia |
| 128 | ER-MD | 9,512 | 209,482 | 10 | NetRepo | Chemical |
| 129 | ER-1_3_20 | 1,017 | 206,432 | N/A | NetRepo | Synthetic-ER |
| 130 | soc-wiki-elec | 7,118 | 201,564 | N/A | NetRepo | Social Networks |
| 131 | soc-epinions | 26,588 | 200,240 | N/A | NetRepo | Social Networks |
| 132 | DeezerEurope | 28,281 | 185,504 | 2 | PyG | Friendship |
| 133 | ca-CondMat | 21,363 | 182,572 | N/A | NetRepo | Collaboration |
| 134 | LINKX-amherst41 | 2,235 | 181,908 | 2 | PyG | Friendship |
| 135 | socfb-Oberlin44 | 2,920 | 179,824 | N/A | NetRepo | Social Networks |
| 136 | econ-orani678 | 2,529 | 173,747 | N/A | NetRepo | Economic Networks |
| 137 | tech-internet-as | 40,164 | 170,246 | N/A | NetRepo | Technology |
| 138 | bio-DR-CX | 3,289 | 169,880 | N/A | NetRepo | Biological Networks |
| 139 | Coauthor-CS | 18,333 | 163,788 | 15 | PyG | Coauthor |
| 140 | ER-1_25_20 | 886 | 157,146 | N/A | NetRepo | Synthetic-ER |
| 141 | HeterophilousGraph-Questions | 48,921 | 153,540 | 2 | PyG | Interaction |
| 142 | bio-DM-CX | 4,040 | 153,434 | N/A | NetRepo | Biological Networks |
| 143 | Entities-MUTAG | 23,644 | 148,454 | 2 | PyG | Knowledgebase |
| 144 | copresence-SFHH | 403 | 147,114 | N/A | NetRepo | Proximity |
| 145 | rec-movielens-tag-movies-10m | 16,528 | 142,148 | N/A | NetRepo | Recommendation |
| 146 | scc_fb-forum | 488 | 142,022 | N/A | NetRepo | Temporal Reachability |
| 147 | BZR-MD | 6,519 | 137,734 | 8 | NetRepo | Chemical |
| 148 | ia-frwikinews-user-edits | 25,042 | 137,354 | N/A | NetRepo | Interaction |
| 149 | scc_retweet | 1,206 | 131,980 | N/A | NetRepo | Temporal Reachability |
| 150 | ER-1_6_20 | 803 | 128,654 | N/A | NetRepo | Synthetic-ER |
| 151 | ER-1_16_10 | 1,126 | 127,004 | N/A | NetRepo | Synthetic-ER |
| 152 | CitationFull-Cora | 19,793 | 126,842 | 70 | PyG | Citation |
| 153 | bio-SC-HT | 2,084 | 126,054 | N/A | NetRepo | Biological Networks |
| 154 | StochasticBlockModel-3.0 | 1,000 | 123,752 | 4 | PyG | Synthetic-SBM |
| 155 | Twitch-ES | 4,648 | 123,412 | 2 | PyG | Friendship |
| 156 | BA-1_9_60 | 1,056 | 123,060 | N/A | NetRepo | Synthetic-BA |
| 157 | socfb-Swarthmore42 | 1,659 | 122,100 | N/A | NetRepo | Social Networks |
| 158 | tech-WHOIS | 7,476 | 113,886 | N/A | NetRepo | Technology |
| 159 | rec-movielens-user-movies-10m | 7,601 | 110,779 | N/A | NetRepo | Recommendation |
| 160 | email-EU | 32,430 | 108,794 | N/A | NetRepo | Email |
| 161 | StochasticBlockModel-2.5 | 1,000 | 107,416 | 4 | PyG | Synthetic-SBM |
| 162 | bio-CE-GN | 2,220 | 107,366 | N/A | NetRepo | Biological Networks |
| 163 | tech-as-caida2007 | 26,475 | 106,762 | N/A | NetRepo | Technology |
| 164 | CitationFull-DBLP | 17,716 | 105,734 | 4 | PyG | Citation |
| 165 | CL-10000-1d7-trial3 | 9,267 | 105,485 | N/A | NetRepo | Synthetic-CL |
| 166 | cit-DBLP | 12,591 | 99,255 | N/A | NetRepo | Citation |
| 167 | soc-anybeat | 12,645 | 98,264 | N/A | NetRepo | Social Networks |
| 168 | KPGM-log12-16-trial3 | 3,324 | 97,110 | N/A | NetRepo | Synthetic-KPGM |
| 169 | bio-CE-PG | 1,871 | 95,508 | N/A | NetRepo | Biological Networks |
| 170 | web-indochina-2004 | 11,358 | 95,212 | N/A | NetRepo | Web Graphs |
| 171 | HeterophilousGraph-Amazon-ratings | 24,492 | 93,050 | 5 | PyG | Co-Purchase |
| 172 | BA-1_6_60 | 803 | 92,700 | N/A | NetRepo | Synthetic-BA |
| 173 | econ-beaflw | 502 | 90,202 | N/A | NetRepo | Economic Networks |
| 174 | StochasticBlockModel-2.0 | 1,000 | 89,892 | 4 | PyG | Synthetic-SBM |
| 175 | BA-1_18_40 | 1,141 | 89,640 | N/A | NetRepo | Synthetic-BA |
| 176 | CitationFull-PubMed | 19,717 | 88,648 | 3 | PyG | Citation |
| 177 | AttributedGraph-Facebook | 4,039 | 88,234 | 193 | PyG | Social Networks |
| 178 | CL-10000-1d8-trial3 | 9,251 | 87,601 | N/A | NetRepo | Synthetic-CL |
| 179 | econ-beacxc | 492 | 84,754 | N/A | NetRepo | Economic Networks |
| 180 | econ-mbeacxc | 487 | 83,776 | N/A | NetRepo | Economic Networks |
| 181 | econ-mbeaflw | 487 | 83,776 | N/A | NetRepo | Economic Networks |
| 182 | soc-advogato | 6,551 | 82,859 | N/A | NetRepo | Social Networks |
| 183 | BA-1_3_40 | 1,017 | 79,720 | N/A | NetRepo | Synthetic-BA |
| 184 | econ-beause | 507 | 79,254 | N/A | NetRepo | Economic Networks |
| 185 | Twitch-RU | 4,385 | 78,993 | 2 | PyG | Friendship |

*Continued on the next page*

| Id | Graph | # Nodes | # Edges | # Node Classes | Data Source | Domain |
|---|---|---|---|---|---|---|
| 186 | bio-HS-LC | 4,227 | 78,968 | N/A | NetRepo | Biological Networks |
| 187 | soc-gplus | 23,628 | 78,388 | N/A | NetRepo | Social Networks |
| 188 | ia-escorts-dynamic | 10,106 | 78,040 | N/A | NetRepo | Interaction |
| 189 | Twitch-EN | 7,126 | 77,774 | 2 | PyG | Friendship |
| 190 | RandomPartitionGraph-hr0.5-ad15 | 5,000 | 75,702 | 10 | PyG | Synthetic-RandPart |
| 191 | KPGM-log12-12-trial2 | 3,214 | 75,682 | N/A | NetRepo | Synthetic-KPGM |
| 192 | RandomPartitionGraph-hr0.7-ad15 | 5,000 | 75,042 | 10 | PyG | Synthetic-RandPart |
| 193 | RandomPartitionGraph-hr0.1-ad15 | 5,000 | 75,026 | 10 | PyG | Synthetic-RandPart |
| 194 | RandomPartitionGraph-hr0.3-ad15 | 5,000 | 74,978 | 10 | PyG | Synthetic-RandPart |
| 195 | web-spam | 4,767 | 74,750 | N/A | NetRepo | Web Graphs |
| 196 | StochasticBlockModel-1.5 | 1,000 | 74,218 | 4 | PyG | Synthetic-SBM |
| 197 | CL-10000-1d9-trial1 | 9,177 | 73,296 | N/A | NetRepo | Synthetic-CL |
| 198 | econ-mbeause | 492 | 72,818 | N/A | NetRepo | Economic Networks |
| 199 | bio-SC-CC | 2,223 | 69,758 | N/A | NetRepo | Biological Networks |
| 200 | ER-3_25_5 | 52,336 | 69,246 | N/A | NetRepo | Synthetic-ER |
| 201 | bio-SC-GT | 1,716 | 67,974 | N/A | NetRepo | Biological Networks |
| 202 | DHFR | 32,075 | 67,352 | 9 | NetRepo | Chemical |
| 203 | AIDS | 31,385 | 64,780 | 38 | NetRepo | Biological Networks |
| 204 | Twitch-PT | 1,912 | 64,510 | 2 | PyG | Friendship |
| 205 | web-wiki-chameleon | 2,277 | 62,792 | N/A | NetRepo | Web Graphs |
| 206 | CL-10000-2d0-trial1 | 9,130 | 62,615 | N/A | NetRepo | Synthetic-CL |
| 207 | soc-political-retweet | 18,470 | 61,157 | 2 | NetRepo | Retweet |
| 208 | MixHopSynthetic-Homophily-0.7 | 5,000 | 59,596 | 10 | PyG | Synthetic-Others |
| 209 | MixHopSynthetic-Homophily-0.3 | 5,000 | 59,596 | 10 | PyG | Synthetic-Others |
| 210 | MixHopSynthetic-Homophily-0.6 | 5,000 | 59,596 | 10 | PyG | Synthetic-Others |
| 211 | MixHopSynthetic-Homophily-0.8 | 5,000 | 59,596 | 10 | PyG | Synthetic-Others |
| 212 | MixHopSynthetic-Homophily-0.9 | 5,000 | 59,596 | 10 | PyG | Synthetic-Others |
| 213 | MixHopSynthetic-Homophily-0.1 | 5,000 | 59,596 | 10 | PyG | Synthetic-Others |
| 214 | MixHopSynthetic-Homophily-0.0 | 5,000 | 59,596 | 10 | PyG | Synthetic-Others |
| 215 | MixHopSynthetic-Homophily-0.4 | 5,000 | 59,596 | 10 | PyG | Synthetic-Others |
| 216 | MixHopSynthetic-Homophily-0.5 | 5,000 | 59,596 | 10 | PyG | Synthetic-Others |
| 217 | MixHopSynthetic-Homophily-0.2 | 5,000 | 59,596 | 10 | PyG | Synthetic-Others |
| 218 | CL-10000-2d1-trial2 | 9,078 | 59,026 | N/A | NetRepo | Synthetic-CL |
| 219 | Entities-AIFB | 8,285 | 58,086 | 4 | PyG | Knowledgebase |
| 220 | LastFMAsia | 7,624 | 55,612 | 18 | PyG | Friendship |
| 221 | KPGM-log12-8-trial3 | 2,968 | 53,510 | N/A | NetRepo | Synthetic-KPGM |
| 222 | reality-call | 27,045 | 52,050 | 2 | NetRepo | Phone Call Networks |
| 223 | web-webbase-2001 | 16,062 | 51,186 | N/A | NetRepo | Web Graphs |
| 224 | econ-poli-large | 15,575 | 50,511 | N/A | NetRepo | Economic Networks |
| 225 | RandomPartitionGraph-hr0.7-ad10 | 5,000 | 49,974 | 10 | PyG | Synthetic-RandPart |
| 226 | RandomPartitionGraph-hr0.5-ad10 | 5,000 | 49,688 | 10 | PyG | Synthetic-RandPart |
| 227 | RandomPartitionGraph-hr0.3-ad10 | 5,000 | 49,548 | 10 | PyG | Synthetic-RandPart |
| 228 | RandomPartitionGraph-hr0.1-ad10 | 5,000 | 49,434 | 10 | PyG | Synthetic-RandPart |
| 229 | tech-pgp | 10,680 | 48,632 | N/A | NetRepo | Technology |
| 230 | scc_retweet-crawl | 17,151 | 48,030 | N/A | NetRepo | Temporal Reachability |
| 231 | ER-2_7_5 | 9,583 | 46,642 | N/A | NetRepo | Synthetic-ER |
| 232 | BA-1_10_60-L5 | 804 | 46,410 | 5 | NetRepo | Synthetic-BA |
| 233 | CL-10K-1d8-L5 | 10,000 | 44,896 | 5 | NetRepo | Synthetic-CL |
| 234 | MSRC-9 | 8,968 | 43,288 | 10 | NetRepo | Computer Vision |
| 235 | bio-SC-LC | 2,004 | 40,904 | N/A | NetRepo | Biological Networks |
| 236 | MSRC-21C | 8,418 | 40,380 | 21 | NetRepo | Computer Vision |
| 237 | ER-3_16_5 | 32,358 | 40,346 | N/A | NetRepo | Synthetic-ER |
| 238 | StochasticBlockModel-0.5 | 1,000 | 40,020 | 4 | PyG | Synthetic-SBM |
| 239 | StochasticBlockModel-1.0 | 1,000 | 40,020 | 4 | PyG | Synthetic-SBM |
| 240 | HeterophilousGraph-Minesweeper | 10,000 | 39,402 | 2 | PyG | Synthetic-Others |
| 241 | web-BerkStan | 12,305 | 39,000 | N/A | NetRepo | Web Graphs |
| 242 | LINKX-reed98 | 962 | 37,624 | 2 | PyG | Friendship |
| 243 | WikipediaNetwork-Chameleon | 2,277 | 36,101 | 5 | PyG | Wikipedia |
| 244 | IMDB | 11,616 | 34,212 | 3 | PyG | Knowledgebase |
| 245 | ER-1_14_5 | 829 | 34,184 | N/A | NetRepo | Synthetic-ER |
| 246 | copresence-InVS15 | 219 | 33,450 | N/A | NetRepo | Proximity |
| 247 | socfb-Caltech | 769 | 33,312 | N/A | NetRepo | Social Networks |
| 248 | soc-hamsterster | 2,426 | 33,260 | N/A | NetRepo | Social Networks |
| 249 | HeterophilousGraph-Roman-empire | 22,662 | 32,927 | 18 | PyG | Wikipedia |
| 250 | bn-mouse-brain1 | 213 | 32,331 | N/A | NetRepo | Brain Networks |
| 251 | inf-openflights | 2,939 | 31,354 | N/A | NetRepo | Infrastructure |
| 252 | BZR | 14,479 | 31,070 | 10 | NetRepo | Chemical |
| 253 | Actor | 7,600 | 30,019 | 5 | PyG | Wikipedia |
| 254 | SW-10000-6-0d3-L5 | 10,000 | 30,000 | 5 | NetRepo | Synthetic-Others |
| 255 | SW-10000-6-0d3-L2 | 10,000 | 30,000 | 2 | NetRepo | Synthetic-Others |
| 256 | soc-sign-bitcoinalpha | 3,783 | 28,248 | N/A | NetRepo | Social Networks |
| 257 | bio-HS-HT | 2,570 | 27,382 | N/A | NetRepo | Biological Networks |
| 258 | ca-GrQc | 4,158 | 26,844 | N/A | NetRepo | Collaboration |
| 259 | EmailEUCore | 1,005 | 25,571 | 42 | PyG | Email |
| 260 | bio-grid-fission-yeast | 2,026 | 25,274 | N/A | NetRepo | Biological Networks |

| Id | Graph | # Nodes | # Edges | # Node Classes | Data Source | Domain |
|---|---|---|---|---|---|---|
| 261 | RandomPartitionGraph-hr0.5-ad5 | 5,000 | 25,176 | 10 | PyG | Synthetic-RandPart |
| 262 | RandomPartitionGraph-hr0.7-ad5 | 5,000 | 25,056 | 10 | PyG | Synthetic-RandPart |
| 263 | RandomPartitionGraph-hr0.1-ad5 | 5,000 | 24,754 | 10 | PyG | Synthetic-RandPart |
| 264 | RandomPartitionGraph-hr0.3-ad5 | 5,000 | 24,704 | 10 | PyG | Synthetic-RandPart |
| 265 | ca-DBLP-kang | 2,879 | 22,652 | N/A | NetRepo | Collaboration |
| 266 | power-bcspwr10 | 5,300 | 21,842 | N/A | NetRepo | Power Networks |
| 267 | KPGM-log10-16-trial2 | 883 | 21,148 | N/A | NetRepo | Synthetic-KPGM |
| 268 | email-dnc-corecipient | 906 | 20,858 | N/A | NetRepo | Email |
| 269 | BA-1_8_10 | 1,040 | 20,690 | N/A | NetRepo | Synthetic-BA |
| 270 | rt_lolgop | 9,765 | 20,150 | N/A | NetRepo | Retweet |
| 271 | scc_enron-only | 146 | 19,656 | N/A | NetRepo | Temporal Reachability |
| 272 | rt_barackobama | 9,631 | 19,547 | N/A | NetRepo | Retweet |
| 273 | rt_justinbieber | 9,405 | 19,167 | N/A | NetRepo | Retweet |
| 274 | SFHH-conf-sensor | 403 | 19,130 | N/A | NetRepo | Proximity |
| 275 | PolBlogs | 1,490 | 19,025 | 2 | PyG | Web Graphs |
| 276 | power-eris1176 | 1,176 | 18,552 | N/A | NetRepo | Power Networks |
| 277 | AttributedGraph-Wiki | 2,405 | 17,981 | 17 | PyG | Wikipedia |
| 278 | bn-fly-drosophila-medulla1 | 1,781 | 17,927 | N/A | NetRepo | Brain Networks |
| 279 | web-EPA | 4,271 | 17,818 | N/A | NetRepo | Web Graphs |
| 280 | BA-1_17_10 | 895 | 17,790 | N/A | NetRepo | Synthetic-BA |
| 281 | rt_gmanews | 8,373 | 17,438 | N/A | NetRepo | Retweet |
| 282 | BA-1_1_10 | 862 | 17,130 | N/A | NetRepo | Synthetic-BA |
| 283 | rt_mittromney | 7,974 | 17,074 | N/A | NetRepo | Retweet |
| 284 | KPGM-log10-12-trial1 | 845 | 16,934 | N/A | NetRepo | Synthetic-KPGM |
| 285 | primary-school-proximity | 242 | 16,634 | N/A | NetRepo | Proximity |
| 286 | BA-1_12_10 | 827 | 16,430 | N/A | NetRepo | Synthetic-BA |
| 287 | CitationFull-CoraML | 2,995 | 16,316 | 7 | PyG | Citation |
| 288 | rt_ksa | 7,302 | 16,216 | N/A | NetRepo | Retweet |
| 289 | rt_onedirection | 7,987 | 16,203 | N/A | NetRepo | Retweet |
| 290 | rt_saudi | 7,252 | 16,121 | N/A | NetRepo | Retweet |
| 291 | ia-reality | 6,809 | 15,360 | N/A | NetRepo | Interaction |
| 292 | econ-mahindas | 1,258 | 15,132 | N/A | NetRepo | Economic Networks |
| 293 | ca-Erdos992 | 5,094 | 15,030 | N/A | NetRepo | Collaboration |
| 294 | rt_dash | 6,288 | 14,870 | N/A | NetRepo | Retweet |
| 295 | DD21 | 5,748 | 14,267 | 21 | NetRepo | Misc |
| 296 | rt_alwefaq | 4,171 | 14,122 | N/A | NetRepo | Retweet |
| 297 | Airports-USA | 1,190 | 13,599 | 4 | PyG | Flight |
| 298 | tech-routers-rf | 2,113 | 13,264 | N/A | NetRepo | Technology |
| 299 | power-US-Grid | 4,941 | 13,188 | N/A | NetRepo | Power Networks |
| 300 | Peking-1 | 3,341 | 13,150 | 190 | NetRepo | Social Networks |
| 301 | web-edu | 3,031 | 12,948 | N/A | NetRepo | Web Graphs |
| 302 | rt_uae | 5,248 | 12,772 | N/A | NetRepo | Retweet |
| 303 | rt_oman | 4,904 | 12,456 | N/A | NetRepo | Retweet |
| 304 | scc_infect-hyper | 113 | 12,444 | N/A | NetRepo | Temporal Reachability |
| 305 | econ-poli | 4,008 | 12,246 | N/A | NetRepo | Economic Networks |
| 306 | KPGM-log10-8-trial2 | 796 | 12,080 | N/A | NetRepo | Synthetic-KPGM |
| 307 | rt_p2 | 4,902 | 12,034 | N/A | NetRepo | Retweet |
| 308 | contacts-prox-high-school-2013 | 327 | 11,636 | N/A | NetRepo | Proximity |
| 309 | rt_gop | 4,687 | 11,058 | N/A | NetRepo | Retweet |
| 310 | rt_tcot | 4,547 | 11,004 | N/A | NetRepo | Retweet |
| 311 | email-univ | 1,133 | 10,902 | N/A | NetRepo | Email |
| 312 | CitationFull-CiteSeer | 4,230 | 10,674 | 6 | PyG | Citation |
| 313 | ca-cora | 2,708 | 10,556 | N/A | NetRepo | Collaboration |
| 314 | PTC-FR | 5,110 | 10,532 | 19 | NetRepo | Chemical |
| 315 | DD6 | 4,152 | 10,320 | 20 | NetRepo | Misc |
| 316 | PTC-FM | 4,925 | 10,110 | 18 | NetRepo | Chemical |
| 317 | PTC-MR | 4,915 | 10,108 | 18 | NetRepo | Chemical |
| 318 | CL-1000-1d7-trial2 | 932 | 9,755 | N/A | NetRepo | Synthetic-CL |
| 319 | PTC-MM | 4,695 | 9,624 | 20 | NetRepo | Chemical |
| 320 | bio-DM-HT | 2,989 | 9,320 | N/A | NetRepo | Biological Networks |
| 321 | CL-1000-1d7-trial1 | 928 | 9,279 | N/A | NetRepo | Synthetic-CL |
| 322 | rt_islam | 4,497 | 9,232 | N/A | NetRepo | Retweet |
| 323 | scc_rt_lolgop | 273 | 9,020 | N/A | NetRepo | Temporal Reachability |
| 324 | rt_tlot | 3,665 | 8,949 | N/A | NetRepo | Retweet |
| 325 | rt_lebanon | 3,961 | 8,871 | N/A | NetRepo | Retweet |
| 326 | email-dnc-leak | 1,891 | 8,849 | N/A | NetRepo | Email |
| 327 | TerroristRel | 881 | 8,592 | 2 | NetRepo | Collaboration |
| 328 | bio-SC-TS | 636 | 7,918 | N/A | NetRepo | Biological Networks |
| 329 | biogrid-human | 2,005 | 7,918 | N/A | NetRepo | Biological Networks |
| 330 | rt_occupy | 3,225 | 7,883 | N/A | NetRepo | Retweet |
| 331 | copresence-InVS13 | 95 | 7,830 | N/A | NetRepo | Proximity |
| 332 | rt_damascus | 3,052 | 7,738 | N/A | NetRepo | Retweet |
| 333 | rt_occupywallstnyc | 3,609 | 7,663 | N/A | NetRepo | Retweet |
| 334 | biogrid-worm | 1,930 | 7,152 | N/A | NetRepo | Biological Networks |
| 335 | road-minnesota | 2,642 | 6,606 | N/A | NetRepo | Road |

| Id | Graph | # Nodes | # Edges | # Node Classes | Data Source | Domain |
|---|---|---|---|---|---|---|
| 336 | power-bcspwr09 | 1,723 | 6,511 | N/A | NetRepo | Power Networks |
| 337 | email-radoslaw | 167 | 6,501 | N/A | NetRepo | Email |
| 338 | bio-CE-GT | 924 | 6,478 | N/A | NetRepo | Biological Networks |
| 339 | bn-macaque-rhesus-brain1 | 242 | 6,108 | N/A | NetRepo | Brain Networks |
| 340 | CL-1000-2d0-trial3 | 916 | 6,004 | N/A | NetRepo | Synthetic-CL |
| 341 | Airports-Europe | 399 | 5,995 | 4 | PyG | Flight |
| 342 | bio-CE-HT | 2,617 | 5,970 | N/A | NetRepo | Biological Networks |
| 343 | CL-1000-2d0-trial2 | 899 | 5,861 | N/A | NetRepo | Synthetic-CL |
| 344 | soc-wiki-Vote | 889 | 5,828 | N/A | NetRepo | Social Networks |
| 345 | rt_assad | 2,139 | 5,574 | N/A | NetRepo | Retweet |
| 346 | web-google | 1,299 | 5,546 | N/A | NetRepo | Web Graphs |
| 347 | infect-dublin | 410 | 5,530 | N/A | NetRepo | Proximity |
| 348 | CL-1000-2d1-trial2 | 911 | 5,457 | N/A | NetRepo | Synthetic-CL |
| 349 | AttributedGraph-Cora | 2,708 | 5,429 | 7 | PyG | Citation |
| 350 | econ-wm1 | 260 | 4,943 | N/A | NetRepo | Economic Networks |
| 351 | rt_voteonedirection | 2,280 | 4,928 | N/A | NetRepo | Retweet |
| 352 | econ-wm3 | 259 | 4,918 | N/A | NetRepo | Economic Networks |
| 353 | econ-wm2 | 259 | 4,908 | N/A | NetRepo | Economic Networks |
| 354 | AttributedGraph-CiteSeer | 3,312 | 4,715 | 6 | PyG | Citation |
| 355 | web-polblogs | 643 | 4,560 | N/A | NetRepo | Web Graphs |
| 356 | bn-macaque-rhesus-interareal-cortical2 | 93 | 4,524 | N/A | NetRepo | Brain Networks |
| 357 | infect-hyper | 113 | 4,392 | N/A | NetRepo | Proximity |
| 358 | eco-foodweb-baydry | 128 | 4,212 | N/A | NetRepo | Ecology |
| 359 | eco-foodweb-baywet | 128 | 4,150 | N/A | NetRepo | Ecology |
| 360 | eco-florida | 128 | 4,150 | N/A | NetRepo | Ecology |
| 361 | power-1138-bus | 1,138 | 4,054 | N/A | NetRepo | Power Networks |
| 362 | KPGM-log8-12-trial3 | 230 | 3,522 | N/A | NetRepo | Synthetic-KPGM |
| 363 | ca-CSphd | 1,882 | 3,480 | N/A | NetRepo | Collaboration |
| 364 | DD242 | 1,284 | 3,303 | 20 | NetRepo | Misc |
| 365 | bio-CE-LC | 1,387 | 3,296 | N/A | NetRepo | Biological Networks |
| 366 | biogrid-mouse | 1,450 | 3,272 | N/A | NetRepo | Biological Networks |
| 367 | power-685-bus | 685 | 3,249 | N/A | NetRepo | Power Networks |
| 368 | bn-mouse-kasthuri-v4 | 1,029 | 3,118 | N/A | NetRepo | Brain Networks |
| 369 | KPGM-log8-10-trial3 | 224 | 3,040 | N/A | NetRepo | Synthetic-KPGM |
| 370 | ia-crime-moreno | 829 | 2,948 | N/A | NetRepo | Interaction |
| 371 | eco-mangwet | 97 | 2,892 | N/A | NetRepo | Ecology |
| 372 | inf-euroroad | 1,174 | 2,834 | N/A | NetRepo | Infrastructure |
| 373 | road-euroroad | 1,174 | 2,834 | N/A | NetRepo | Road |
| 374 | bn-macaque-rhesus-cerebral-cortex1 | 91 | 2,802 | N/A | NetRepo | Brain Networks |
| 375 | copresence-LH10 | 73 | 2,762 | N/A | NetRepo | Proximity |
| 376 | DD_g106 | 574 | 2,710 | N/A | NetRepo | Protein |
| 377 | KPGM-log8-8-trial3 | 215 | 2,606 | N/A | NetRepo | Synthetic-KPGM |
| 378 | road-ChicagoRegional | 1,467 | 2,596 | N/A | NetRepo | Road |
| 379 | power-662-bus | 662 | 2,474 | N/A | NetRepo | Power Networks |
| 380 | DD_g105 | 423 | 2,384 | N/A | NetRepo | Protein |
| 381 | hospital-ward-proximity | 75 | 2,278 | N/A | NetRepo | Proximity |
| 382 | DD_g108 | 483 | 2,274 | N/A | NetRepo | Protein |
| 383 | bio-DM-LC | 658 | 2,258 | N/A | NetRepo | Biological Networks |
| 384 | scc_rt_gmanews | 135 | 2,156 | N/A | NetRepo | Temporal Reachability |
| 385 | biogrid-yeast | 836 | 2,098 | N/A | NetRepo | Biological Networks |
| 386 | rt-twitter-copen | 761 | 2,058 | N/A | NetRepo | Retweet |
| 387 | DD_g100 | 349 | 2,010 | N/A | NetRepo | Protein |
| 388 | DD_g104 | 372 | 1,998 | N/A | NetRepo | Protein |
| 389 | DD_g115 | 336 | 1,892 | N/A | NetRepo | Protein |
| 390 | scc_rt_occupywallstnyc | 127 | 1,862 | N/A | NetRepo | Temporal Reachability |
| 391 | soc-physicians | 241 | 1,846 | N/A | NetRepo | Social Networks |
| 392 | ca-netscience | 379 | 1,828 | N/A | NetRepo | Collaboration |
| 393 | eco-everglades | 69 | 1,765 | N/A | NetRepo | Ecology |
| 394 | biogrid-plant | 523 | 1,676 | N/A | NetRepo | Biological Networks |
| 395 | power-494-bus | 494 | 1,666 | N/A | NetRepo | Power Networks |
| 396 | DD_g1021 | 329 | 1,574 | N/A | NetRepo | Protein |
| 397 | DD_g11 | 312 | 1,522 | N/A | NetRepo | Protein |
| 398 | ia-workplace-contacts | 92 | 1,510 | N/A | NetRepo | Interaction |
| 399 | bn-cat-mixed-species-brain1 | 65 | 1,460 | N/A | NetRepo | Brain Networks |
| 400 | DD_g1022 | 294 | 1,460 | N/A | NetRepo | Protein |
| 401 | DD_g101 | 306 | 1,456 | N/A | NetRepo | Protein |
| 402 | DD_g103 | 265 | 1,294 | N/A | NetRepo | Protein |
| 403 | email-enron-only | 143 | 1,246 | N/A | NetRepo | Email |
| 404 | bn-macaque-rhesus-brain2 | 91 | 1,164 | N/A | NetRepo | Brain Networks |
| 405 | DD_g1006 | 246 | 1,136 | N/A | NetRepo | Protein |
| 406 | Airports-Brazil | 131 | 1,074 | 4 | PyG | Flight |
| 407 | scc_rt_justinbieber | 62 | 884 | N/A | NetRepo | Temporal Reachability |
| 408 | DD_g1000 | 183 | 816 | N/A | NetRepo | Protein |
| 409 | DD_g1017 | 162 | 752 | N/A | NetRepo | Protein |
| 410 | scc_rt_alwefaq | 72 | 710 | N/A | NetRepo | Temporal Reachability |

*Continued on the next page*

| Id | Graph | # Nodes | # Edges | # Node Classes | Data Source | Domain |
|---|---|---|---|---|---|---|
| 411 | DD_g1019 | 131 | 706 | N/A | NetRepo | Protein |
| 412 | eco-stmarks | 54 | 703 | N/A | NetRepo | Ecology |
| 413 | DD_g1030 | 136 | 702 | N/A | NetRepo | Protein |
| 414 | DD_g10 | 146 | 656 | N/A | NetRepo | Protein |
| 415 | internet-industry-partnerships | 219 | 631 | 3 | NetRepo | Collaboration |
| 416 | soc-student-coop | 185 | 622 | N/A | NetRepo | Social Networks |
| 417 | DD_g1016 | 113 | 582 | N/A | NetRepo | Protein |
| 418 | soc-highschool-moreno | 70 | 548 | N/A | NetRepo | Social Networks |
| 419 | DD_g1009 | 129 | 544 | N/A | NetRepo | Protein |
| 420 | webkb-wisc | 265 | 530 | 5 | NetRepo | Web Graphs |
| 421 | WebKB-Wisconsin | 251 | 515 | 5 | PyG | Web Graphs |
| 422 | DD_g1015 | 102 | 488 | N/A | NetRepo | Protein |
| 423 | DD_g1004 | 94 | 460 | N/A | NetRepo | Protein |
| 424 | scc_rt_barackobama | 80 | 452 | N/A | NetRepo | Temporal Reachability |
| 425 | DD_g1027 | 108 | 446 | N/A | NetRepo | Protein |
| 426 | bn-mouse-visual-cortex2 | 193 | 428 | N/A | NetRepo | Brain Networks |
| 427 | DD_g1025 | 88 | 410 | N/A | NetRepo | Protein |
| 428 | WebKB-Texas | 183 | 325 | 5 | PyG | Web Graphs |
| 429 | WebKB-Cornell | 183 | 298 | 5 | PyG | Web Graphs |
| 430 | ENZYMES_g297 | 121 | 298 | N/A | NetRepo | Chemical |
| 431 | ENZYMES_g296 | 125 | 282 | N/A | NetRepo | Chemical |
| 432 | DD_g1028 | 72 | 274 | N/A | NetRepo | Protein |
| 433 | ENZYMES_g118 | 95 | 242 | N/A | NetRepo | Chemical |
| 434 | ENZYMES_g504 | 66 | 240 | N/A | NetRepo | Chemical |
| 435 | DD_g1003 | 53 | 232 | N/A | NetRepo | Protein |
| 436 | ENZYMES_g103 | 59 | 230 | N/A | NetRepo | Chemical |
| 437 | ENZYMES_g594 | 52 | 228 | N/A | NetRepo | Chemical |
| 438 | ENZYMES_g355 | 66 | 224 | N/A | NetRepo | Chemical |
| 439 | NCI1_g3139 | 107 | 224 | N/A | NetRepo | Chemical |
| 440 | NCI1_g1863 | 107 | 222 | N/A | NetRepo | Chemical |
| 441 | ENZYMES_g575 | 51 | 220 | N/A | NetRepo | Chemical |
| 442 | ENZYMES_g526 | 58 | 220 | N/A | NetRepo | Chemical |
| 443 | ENZYMES_g199 | 62 | 216 | N/A | NetRepo | Chemical |
| 444 | ENZYMES_g279 | 60 | 214 | N/A | NetRepo | Chemical |
| 445 | NCI1_g3585 | 105 | 214 | N/A | NetRepo | Chemical |
| 446 | ENZYMES_g527 | 57 | 214 | N/A | NetRepo | Chemical |
| 447 | NCI1_g1677 | 102 | 212 | N/A | NetRepo | Chemical |
| 448 | NCI1_g3711 | 89 | 212 | N/A | NetRepo | Chemical |
| 449 | ENZYMES_g224 | 54 | 210 | N/A | NetRepo | Chemical |
| 450 | NCI1_g3990 | 90 | 210 | N/A | NetRepo | Chemical |
| 451 | ENZYMES_g291 | 62 | 208 | N/A | NetRepo | Chemical |
| 452 | NCI1_g2079 | 88 | 206 | N/A | NetRepo | Chemical |
| 453 | NCI1_g3444 | 93 | 204 | N/A | NetRepo | Chemical |
| 454 | NCI1_g1893 | 96 | 204 | N/A | NetRepo | Chemical |
| 455 | NCI1_g2082 | 86 | 202 | N/A | NetRepo | Chemical |
| 456 | ENZYMES_g598 | 55 | 200 | N/A | NetRepo | Chemical |
| 457 | KarateClub | 34 | 156 | 4 | PyG | Friendship |

# G   Hosting, Licensing, and Maintenance Plan

## G.1   Hosting

The code and data of GLEMOS are hosted at `https://namyongpark.github.io/glemos`.

## G.2   Licensing

**Data License.** Among data included in GLEMOS, performance records (a), splitting of graph data (b) and testbed data (c), and meta-graph features (d) were generated by GLEMOS by processing the graph data (Appendix F.2). They are under the CC BY-NC 4.0 license. Graph data (Appendix F.2) are from two public graph repositories, *i.e.*, Network Repository [19] and PyTorch Geometric [6]. Since all graph data used in this work are from these repositories, please refer to the corresponding repository for the license of graph datasets.

**Code License.** The GLEMOS codebase is under the CC BY-NC 4.0 license.

## G.3   Maintenance

We will maintain and continue to develop GLEMOS for the long term. Specifically, we aim to improve and expand GLEMOS in two aspects, *i.e.*, benchmark data and model selection algorithms.

**Benchmark Data.** We will monitor newly available graphs from various domains, as well as new graph learning (GL) models, and expand GLEMOS with those new graphs and GL models as follows.

- *Performance Records:* We will evaluate new GL models on both new and existing graphs for applicable GL tasks, and evaluate existing GL models on the new graphs as well. Those new results will enrich our collection of performance records.
- *Graph Data Splits:* Node and edge splits to evaluate GL models on the new graphs will be added.
- *Testbed Data Splits:* Testbed data splits (*e.g.*, over the performance matrix) used to evaluate model selection algorithms will be shared.
- *Meta-Graph Features:* We will generate different sets of meta-graph features for the new graphs.

**Model Selection Algorithms.** As we discuss in the main text, there exist multiple directions for future work to improve the effectiveness and generalization capability of GL model selection algorithms. We will continue to monitor the latest advancement of this area, and expand GLEMOS with the state-of-the-art GL model selection algorithms.

## H   Author Statement

We take all responsibilities of the benchmark data. In case of violation of any rights or data licenses, we will take necessary actions, such as revising the problematic data, or removing it from the benchmark.