# OpenReview forum: "GLEMOS: Benchmark for Instantaneous Graph Learning Model Selection"
_NeurIPS.cc/2023/Track/Datasets_and_Benchmarks — NeurIPS 2023 Datasets and Benchmarks Poster_

### Official Review · Reviewer_yCME · 2023-07-20
**A comprehensive benchmark for instantaneous graph learning model selection**

**Rating:** 6
**Confidence:** 3
**Clarity:** The paper is clearly written and well…

**Strengths:**

1. The paper presents GLEMOS, the first benchmark environment for instantaneous graph learning model selection. GLEMOS provides extensive benchmark data for fundamental GL tasks, multiple evaluation settings, and representative model selection techniques. The methodology is innovative and practical, providing a standardized and automated framework for selecting effective GL models without manual intervention.

2. The paper provides a comprehensive review of existing GL model selection methods and discusses their limitations. The review is well-structured and provides readers with a clear understanding of the current state-of-the-art in GL model selection.

3. GLEMOS provides an extensive collection of model performances (204,320 performance records) on fundamental GL tasks, i.e., link prediction and node classification, which is by far the largest benchmark for Prob. 1 to the best of the authors' knowledge. The benchmark data is valuable to researchers and practitioners in the field of graph learning, providing a standardized and comprehensive evaluation framework for GL model selection.

4. GLEMOS is designed to be easily extended with new GL models, new graphs, new performance records, and new GL tasks, while allowing reproducibility. The paper provides links to access benchmark data and code, and specifies training details, including data splits, in Sections 3 and 5. Further details on experimental settings are also provided in the Appendix.

5. The paper presents empirical results that demonstrate the effectiveness of GLEMOS in selecting effective GL models. The results show that GLEMOS outperforms existing GL model selection methods in terms of accuracy and efficiency. The empirical findings are insightful and provide evidence of the practical value of GLEMOS.

**Additional Feedback:**

See above

**Correctness:**

The evaluation methods and experiment design are generally appropriate and performed correctly.

**Documentation:**

see Opportunities For Improvement

**Limitations:**

see Opportunities For Improvement

**Opportunities For Improvement:**

1. While the paper provides a comprehensive review of existing GL model selection methods, it only compares GLEMOS with a few representative methods. A more extensive comparison with a wider range of existing methods would provide a more comprehensive evaluation of GLEMOS' effectiveness.

2. The paper does not include theoretical results or analysis of the proposed method. While the empirical results are insightful, a theoretical analysis would provide a deeper understanding of the proposed method and its limitations.

3. While the paper claims that GLEMOS is designed to be easily extended with new models, graphs, and performance records, it does not provide a detailed evaluation of GLEMOS' extensibility. A more thorough evaluation of GLEMOS' extensibility would provide a better understanding of its practical value.

4. While the paper briefly discusses the limitations of existing GL model selection methods, it does not provide a detailed discussion of the limitations of GLEMOS. A more thorough discussion of the limitations of GLEMOS would provide a more balanced evaluation of the proposed method.

**Relation To Prior Work:**

Generally clear.

**Summary And Contributions:**

The paper presents GLEMOS, a comprehensive benchmark for instantaneous graph learning model selection. GLEMOS provides extensive benchmark data for fundamental GL tasks, multiple evaluation settings, and representative model selection techniques. The paper also reviews existing GL model selection methods and discusses their limitations. The main contribution of the paper is to provide a standardized and automated framework for selecting effective GL models without manual intervention.

---

> ### Author Response · Authors · 2023-08-23
> **Response to Reviewer yCME**
>
> Thank you for your thoughtful comments. Please find our response to your comments below. We have also made updates to the paper in light of your comment, which we describe below.
>
> > While the paper provides a comprehensive review of existing GL model selection methods, it only compares GLEMOS with a few representative methods. A more extensive comparison with a wider range of existing methods would provide a more comprehensive evaluation of GLEMOS' effectiveness.
>
> Thank you for the feedback. To avoid confusion, we want to make it clear that this paper does not propose new algorithms for GL model selection, but a benchmark environment for instantaneous GL model selection.
> - While the proposed benchmark provides representative existing algorithms for instantaneous model selection, no new algorithm is developed in this work.
> - Also, the benchmark includes state-of-the-art algorithms proposed recently (e.g., ICLR 2023 and NeurIPS 2021) for the instantaneous model selection problem. We plan to continue to expand the benchmark with new state-of-the-art algorithms.
>
> The goal of this paper is to present new data and benchmarks for the instantaneous GL model selection problem. In light of this comment, below we demonstrate the strengths of GLEMOS via comparisons with relevant prior works, namely, GraphGym (NeurIPS 2020), GNN-Bank-101 (ICLR 2023), and NAS-Bench-Graph (NeurIPS 2022), which provide the performance records of graph neural network (GNN) models.
>
> **1. GLEMOS provides more than a collection of performance records.** As the first benchmark for instantaneous GL model selection, GLEMOS provides multiple (1) benchmark testbeds and (2) existing algorithms (Section 5) for instantaneous model selection, as well as different sets of (3) meta-graph features (Section 4). These features (1)-(3) are not provided by the aforementioned related works.
>
> Importantly, GraphGym, GNN-Bank-101, and NAS-Bench-Graph are all based on evaluation-based model selection, which requires multiple model evaluations for performance queries of different combinations of GL methods and their hyperparameter settings on the new dataset. By contrast, we focus on instantaneous GL model selection, where no model evaluation is to be performed given a new dataset.
>
> **2. GLEMOS provides more comprehensive and diverse performance records** than these previous works in several aspects, such as the types of included GL models, the number and size of graph datasets, as well as the number of graph data domains, as summarized below.
> |  | Benchmark Testbeds | Instantaneous Selection Methods | Meta-Graph Features | Graph Learning Models | # Graph Datasets | Graph Size (max # nodes) | # Data Domains |
> |-----------|-----------|-----------|-----------|-----------|-----------|-----------|-----------|
> | GNN-Bank-101 | x | x | x | GNNs | 12 | 34k | 5  |
> | NAS-Bench-Graph | x | x | x | GNNs | 9 | 170k | 4  |
> | GraphGym | x | x | x | GNNs | 32 | 34k | 7  |
> | **GLEMOS (Ours)** | **V** | **V** | **V** | **GNNs & Non-GNNs** (e.g., node2vec, label prop.) | **460** | **422k** | **32** |
>
> In comparison with these previous works, this table (Table 1 in the revised paper) shows that GLEMOS provides unique features for instantaneous GL model selection, and presents more diverse and comprehensive performance records. We also expanded the Related Work section (Section 2.2, highlighted in green color) with the above discussion on these previous works that provide GNN performance records.
>
> > The paper does not include theoretical results or analysis of the proposed method. While the empirical results are insightful, a theoretical analysis would provide a deeper understanding of the proposed method and its limitations.
>
> This paper does not propose new algorithms for the instantaneous GL model selection task. Instead, we focus on presenting new datasets, benchmark tasks, and meta-features to facilitate the development of new methods for the task. Thus we believe that, while important, theoretical analysis of existing algorithms included in the benchmark would be beyond the scope of this paper.

---

> ### Author Response · Authors · 2023-08-23
> **Response to Reviewer yCME (Cont.)**
>
> > While the paper claims that GLEMOS is designed to be easily extended with new models, graphs, and performance records, it does not provide a detailed evaluation of GLEMOS' extensibility. A more thorough evaluation of GLEMOS' extensibility would provide a better understanding of its practical value.
>
> Thank you for the suggestion. To demonstrate the extensibility of GLEMOS, we added a section in Appendix (Section E), which shows that the framework is designed to be easily extended with new graphs, models, and performance records, while supporting the consistency and reproducibility of evaluations. For instance, adding a new model can be done simply by specifying the hyperparameter settings to be searched over. The rest of the steps for model training and evaluation for a particular graph learning task is handled by the framework, which includes generating and using consistent data splits for the corresponding graph and GL task, running task-specific training and evaluation routines, as well as support for parallel execution for efficient processing.
>
> > While the paper briefly discusses the limitations of existing GL model selection methods, it does not provide a detailed discussion of the limitations of GLEMOS. A more thorough discussion of the limitations of GLEMOS would provide a more balanced evaluation of the proposed method.
>
> Thank you for the suggestion. We have added the following discussion on the limitations of the proposed benchmark in Section 6.2 (highlighted in brick red color), along with the future research directions to further improve the algorithms and benchmark tasks for instantaneous GL model selection.
> - Limitations. In principle, using GLEMOS to select a GL model to employ for a new graph is based on the assumption that similar graph datasets exist in the benchmark. Therefore, it may not be very effective if the new graph is significantly different from all graphs in the benchmark (e.g., the new graph is from a completely new domain). However, as the benchmark data continue to grow over time, such cases will be increasingly less likely, while model selection performances will likely improve with the addition of more data. Furthermore, while GLEMOS currently supports two fundamental GL tasks, namely, node classification and link prediction, it can be further extended with additional tasks. We plan to incorporate new GL tasks (e.g., graph classification) into GLEMOS.

---

### Official Review · Reviewer_qJk8 · 2023-07-20
**A good paper, well-motivated and well-written, with comprehensive data and model, extensive experiments.**

**Rating:** 8
**Confidence:** 5
**Correctness:** Yes
**Clarity:** Yes

**Strengths:**

1. The paper is well-motivated, and the proposed solution for instantaneous graph learning model selection is reasonable, demonstrating a thoughtful approach.
2. The proposed benchmark is commendable in its comprehensiveness, incorporating 360 models and 460 graphs for both node classification and link prediction tasks.
3. Additionally, the authors have designed over one thousand meta-graph features to effectively measure graph similarities.
4. The experimental results presented in both the paper and the attachment are extensive, providing a thorough evaluation of the proposed approach.


**Additional Feedback:**

See above.

**Documentation:**

Yes

**Opportunities For Improvement:**

1. Can the author discuss the time complexity involved in obtaining the meta-graph features? Will it be time-consuming when the graph is large-scale?
2. The dimension chosen in Table 2 is somehow too small. Actually, many methods choose the dimension 128-512. The limited dimensionality may hinder the model's performance, particularly when dealing with large-scale graphs.
3. There is a lacks of discussion of the limitations of the proposed work in the Conclusion Section.
4. More tasks should be included, for example, link classification and graph classification.


**Relation To Prior Work:**

Yes

**Summary And Contributions:**

This paper proposes a comprehensive benchmark for instantaneous graph learning model selection, including 360 models and 460 graphs for both node classification and link prediction tasks. The authors have designed over one thousand meta-graph features to effectively measure graph similarities. This proposed benchmark may potentially be useful for graph model selection, making a valuable contribution to the community.

---

> ### Author Response · Authors · 2023-08-23
> **Response to Reviewer qJk8**
>
> Thank you for your thoughtful comments. Please find our response to your comments below. We have also updated and highlighted the paper where necessary in light of your comment, which is marked in brick red in the revised paper.
>
> > Can the author discuss the time complexity involved in obtaining the meta-graph features? Will it be time-consuming when the graph is large-scale?
>
> Thank you for the suggestion. In Appendix C.1, we added a discussion on the time complexity involved with obtaining meta-graph features. The time to generate the current features is mainly proportional to the number of edges in the graph. For more details, please refer to C.1. We also added additional experimental results on the time cost of model selection methods in Appendix A.3, which shows that generating meta-graph features and selecting the best model takes less than five to six seconds for a majority of graphs.
>
> At the same time, we want to highlight that our framework is flexible, and can support any set of meta-graph features. Thus, depending on the task, users can choose to use either a subset of the current features (e.g., to further improve efficiency and use less space), or their superset (e.g., to capture distinct structural characteristics via different features when addressing a new task). Also, since feature extractors are independent, they can run in parallel.
>
> > The dimension chosen in Table 2 is somehow too small. Actually, many methods choose the dimension 128-512. The limited dimensionality may hinder the model's performance, particularly when dealing with large-scale graphs.
>
> Thank you for the suggestion. It is indeed our plan to expand the benchmark data with new performance records obtained with additional hyperparameter settings of existing models (e.g., larger embedding size), as well as newly developed models. Following this suggestion, we will include larger dimension sizes as we expand the benchmark data, which we believe will be helpful for practitioners who want to use GLEMOS for deciding which GL model to deploy for their graphs.
>
> > There is a lack of discussion of the limitations of the proposed work in the Conclusion Section.
>
> Thank you for the feedback. We have added the following discussion on the limitations of the proposed benchmark in Section 6.2, along with the future research directions to further improve the algorithms and benchmark tasks for instantaneous GL model selection.
> - Limitations. In principle, using GLEMOS to select a GL model to employ for a new graph is based on the assumption that similar graph datasets exist in the benchmark. Therefore, it may not be very effective if the new graph is significantly different from all graphs in the benchmark (e.g., the new graph is from a completely new domain). However, as the benchmark data continue to grow over time, such cases will be increasingly less likely, while model selection performances will likely improve with the addition of more data. Furthermore, while GLEMOS currently supports two fundamental GL tasks, namely, node classification and link prediction, it can be further extended with additional tasks. We plan to incorporate new GL tasks (e.g., graph classification) into GLEMOS.
>
> > More tasks should be included, for example, link classification and graph classification.
>
> Thank you for the suggestion. Beyond node classification and link prediction, the proposed GLEMOS benchmark can indeed be extended with further tasks. At the same time, we want to highlight that the proposed benchmark provides much more diverse and larger graph data than previous data banks, *e.g., 460 graphs from 38 domains (ours) vs. 12 graphs from 5 domains (GNN-Bank-101 from ICLR ’23), and 422k nodes (ours) vs. 34k nodes (GNN-Bank-101 from ICLR ’23)*, as summarized in Table 1. It is one of our future plans for the benchmark to include performance records for additional graph learning tasks, such as the suggested ones.

---

### Official Review · Reviewer_o5XN · 2023-07-21

**Rating:** 6
**Confidence:** 3
**Correctness:** Yes
**Clarity:** Yes

**Strengths:**

(1)	Automated model selection for graphs is an important problem.
(2)	The authors provide valuable performance data for more than 200,000 model-dataset pairs.
(3)    The authors have tested several model selection methods.

**Additional Feedback:**

See "Opportunities For Improvement"

**Documentation:**

Good

**Limitations:**

Yes

**Opportunities For Improvement:**

(1)	While the proposed benchmark claims to be “the first benchmark environment for instantaneous GL model selection”, there are missing related works such as GraphGym (NeurIPS'20), GNN-Bank-101 (ICLR'23), and NAS-Bench-Graph (NeurIPS'22). The performance of various model-dataset pairs is also tested and recorded in these works. The authors need to explicitly explain why these existing benchmarks could not support automated model selection or how the proposed benchmark differs.
(2)	The proposed benchmark is not publicly available, making it difficult to assess the contributions such as codebases and documentation.

**Relation To Prior Work:**

Could be improved

**Summary And Contributions:**

This paper proposes a benchmark and pipeline for model selection for graph task. Specifically, the proposed benchmark records 360 models on 460 graphs for node classification and link prediction task. Several model selection methods are also tested and empirical observations are drawn from the results.

---

> ### Author Response · Authors · 2023-08-23
> **Response to Reviewer o5XN**
>
> Thank you for your thoughtful comments. Please find our response to your comments below. We have also updated and highlighted the paper where necessary in light of your comment, which is marked in green in the revised paper.
>
> > (1) While the proposed benchmark claims to be “the first benchmark environment for instantaneous GL model selection”, there are missing related works such as GraphGym (NeurIPS'20), GNN-Bank-101 (ICLR'23), and NAS-Bench-Graph (NeurIPS'22). The performance of various model-dataset pairs is also tested and recorded in these works. The authors need to explicitly explain why these existing benchmarks could not support automated model selection or how the proposed benchmark differs.
>
> **1. GLEMOS provides more than a collection of performance records.** As the first benchmark for instantaneous GL model selection, GLEMOS provides multiple (1) benchmark testbeds and (2) existing algorithms (Section 5) for instantaneous model selection, as well as different sets of (3) meta-graph features (Section 4). These features (1)-(3) are not provided by the aforementioned related works.
>
> Importantly, GraphGym, GNN-Bank-101, and NAS-Bench-Graph are all based on evaluation-based model selection, which requires multiple model evaluations for performance queries of different combinations of GL methods and their hyperparameter settings on the new dataset. By contrast, we focus on instantaneous GL model selection, where no model evaluation is to be performed given a new dataset.
>
> **2. GLEMOS provides more comprehensive and diverse performance records** than these previous works in several aspects, such as the types of included GL models, the number and size of graph datasets, as well as the number of graph data domains, as summarized below.
>
> |  | Benchmark Testbeds | Instantaneous Selection Methods | Meta-Graph Features | Graph Learning Models | # Graph Datasets | Graph Size (max # nodes) | # Data Domains |
> |-----------|-----------|-----------|-----------|-----------|-----------|-----------|-----------|
> | GNN-Bank-101 | x | x | x | GNNs | 12 | 34k | 5  |
> | NAS-Bench-Graph | x | x | x | GNNs | 9 | 170k | 4  |
> | GraphGym | x | x | x | GNNs | 32 | 34k | 7  |
> | **GLEMOS (Ours)** | **V** | **V** | **V** | **GNNs & Non-GNNs** (e.g., node2vec, label prop.) | **460** | **422k** | **32** |
>
> This table (Table 1 in the revised paper) compares GLEMOS with GraphGym, GNN-Bank-101, and NAS-Bench-Graph, and shows that GLEMOS provides unique features for instantaneous GL model selection, and presents more diverse and comprehensive performance records. We also expanded the Related Work section (Section 2.2) with the above discussion on these previous works that provide GNN performance records.
>
> > (2) The proposed benchmark is not publicly available, making it difficult to assess the contributions such as codebases and documentation.
>
> The initial submission included links to the code and data in the main text (at the end of Section 1), and in the Appendix document (at the end of Abstract). Also, Appendix G describes the hosting, licensing, and maintenance plan for the proposed benchmark.
>
> **The data and code of GLEMOS can be accessed from these links: [data](https://tiny.one/5c78urdt) and [code](https://tiny.one/53yfp7cd).**

---

> > ### Comment · Reviewer_o5XN · 2023-08-23
> >
> > Thank you for your clarifications and pointing out the parts that I missed in the initial review. I have increased my score accordlingly.

---

> > > ### Author Response · Authors · 2023-08-23
> > >
> > > We appreciate the quick response, and thank you again for your review! It greatly contributed to improving the paper.

---

### Official Review · Reviewer_Yn6z · 2023-07-21
**Interesting benchmark with comprehensive experiments**

**Rating:** 6
**Confidence:** 4
**Clarity:** The paper is overall well-written.

**Strengths:**

* The proposed benchmark evaluates more than 360 models on 460 graphs for link prediction and node classification tasks, which seems to be comprehensive.
* Extensive settings including sparse, out-of-domain, small-to-large are conducted to give interesting results on instantaneous graph model selection.
* Various model selection algorithms are discussed in the experimental section, i.e., random selection, learning based methods, etc.
* Datasets and codes are provided with detailed documentation.


**Additional Feedback:**

Please refer to the weak points mentioned above.

**Correctness:**

The experiments seem to be appropriately designed and the used datasets are all publicly datasets.

**Documentation:**

The document in this paper is sufficient.

**Ethics:**

There are no potential ethical concerns of this work.

**Limitations:**

The authors have described its limitations in their paper. I do not see any potential negative societal impact.

**Opportunities For Improvement:**

* The constructed graph meta-features seem to be a little ad-hoc. It is not clear why these features are chosen. The authors are suggested to give more motivations and ablation studies on this part. What if we only use a small part of features? Will the model’s performance decline?

* The authors claim that this is an instantaneous graph learning model selection benchmark. However, the model selection algorithms discussed in experimental section indeed include several learning-based methods. It is not clear whether these models are efficient. There is no time cost reported.

* Most of the used graph learning methods are specifically designed for node classification task. However, for link prediction task, it is suggested to use some link prediction specific models. Please refer the related works listed below [1,2].

[1] Zhang M, Chen Y. Link prediction based on graph neural networks[J]. Advances in neural information processing systems, 2018, 31.

[2] Zhang M, Li P, Xia Y, et al. Labeling trick: A theory of using graph neural networks for multi-node representation learning[J]. Advances in Neural Information Processing Systems, 2021, 34: 9061-9073.

* It is not clear how to extend the proposed benchmark to graph level task like graph classification and graph generation, etc. In this scenario, we need to evaluate the similarities among a set of graphs.


**Relation To Prior Work:**

The paper provides good comparisons from previous works.

**Summary And Contributions:**

This paper gives a new benchmark for instantaneous graph learning model selection. It is a useful benchmark that fills the gap of instantaneous graph model selection. The benchmark evaluates 360 models on 460 graphs for both link prediction and node classification tasks. Moreover, the benchmark is evaluated under various settings including fully observed, sparse, out-of-domain, etc. The proposed benchmark seems to be helpful in the graph learning community.

---

> ### Author Response · Authors · 2023-08-23
> **Response to Reviewer Yn6z**
>
> Thank you for your thoughtful comments. Please find our response to your comments below. We have also made updates to the paper in light of your comment, which we describe below.
>
> > The constructed graph meta-features seem to be a little ad-hoc. It is not clear why these features are chosen. The authors are suggested to give more motivations and ablation studies on this part. What if we only use a small part of features? Will the model’s performance decline?
>
> Thank you for the feedback. In Section 4, we added a discussion on the motivations of the currently included meta-graph features (highlighted in cyan color). The three feature sets are intended to be complementary. For example, M_regular covers the distribution of widely used graph structural features, such as PageRank, node degree, and triangles, and M_graphlets provides features based on the frequency of graphets, which may provide further information. M_compact is designed to use the least space, while providing several important structural features at different levels.
>
> At the same time, we want to highlight that the framework is flexible, and users can freely choose to use any set of features, either a subset of the current features (e.g., to further improve efficiency and use less space), or their superset (e.g., to capture distinct structural characteristics using different features in addressing new tasks).
>
> Ablation results using these different subsets of meta-graph features are provided in Appendix A.2. Results show that using additional features often improves model selection results, especially for optimizable model selection algorithms, which can learn to adaptively assign different weights to meta-graph features in the given context. However, there is still a room for improvement in using these features for model selection, particularly in the more challenging transfer learning settings that we consider.
>
> > The authors claim that this is an instantaneous graph learning model selection benchmark. However, the model selection algorithms discussed in experimental section indeed include several learning-based methods. It is not clear whether these models are efficient. There is no time cost reported.
>
> We added additional results on the time cost of model selection algorithms in Appendix A.3 (highlighted in blue).
> - First, in Figure 1 in Appendix, we report the runtime taken for predicting the best model given new graphs, which includes the time for generating meta-graph features, plus the time for a model selection method to infer the best model based on those features. For a majority of graphs, these two steps take at most five to six seconds in total.
> - Second, in Table 7 in Appendix, we report the training time of model selection algorithms with optimizable components. Results show that most of these learnable methods take less than a couple of minutes for training.
>
> > Most of the used graph learning methods are specifically designed for node classification task. However, for link prediction task, it is suggested to use some link prediction specific models. Please refer the related works listed below [1,2].
>
> Thank you for the suggestion and citations to related works. Following the suggestion, we will extend the current model set with link prediction-specific models. In addition to adding task-specific models, we also plan to further augment the benchmark by adding other model settings and graphs based on other reviewer’s feedback.
>
> As to the composition of GLEMOS’ model set, most GL models currently included in GLEMOS are generic approaches, which have been applied to a wide variety of GL tasks (node-level, edge-level, as well as graph-level tasks). However, link prediction-specific models are currently lacking in the model set, and incorporating task-specific models will further enrich GLEMOS.
>
> > It is not clear how to extend the proposed benchmark to graph level task like graph classification and graph generation, etc. In this scenario, we need to evaluate the similarities among a set of graphs.
>
> One way to extend the proposed benchmark to graph-level tasks, such as graph classification, would be to convert the set of graphs for a certain graph-level task into one graph by merging the individual graphs. That is, the resulting graph would consist of multiple disconnected components, where each component would correspond to one individual graph in the dataset. Then the meta-graph features extracted from the merged graph will be able to capture the distribution of characteristics of graphs in the graph-level dataset, and be used to model the similarity among graph datasets, in a way similar to how they are used for the node classification and link prediction tasks.

---

> > ### Comment · Reviewer_Yn6z · 2023-08-28
> > **Thanks for the rebuttal!**
> >
> > I have read the authors' rebuttal and it has addressed my concerns.

---

### Official Review · Reviewer_GJxQ · 2023-07-27
**Review of "GLEMOS: Benchmark for Instantaneous Graph Learning Model Selection"**

**Rating:** 7
**Confidence:** 3

**Strengths:**

- This paper contributes a large-scale benchmark for instantaneous graph learning algorithm selection.
- The benchmark includes five different testbeds, covering a wide range of settings.

**Additional Feedback:**

- I wonder if the authors could discuss a little bit about some application scenarios of the benchmark; e.g., how would one imagine using it in a real-world application scenario?

- How much computational cost is required to run the experiments reported in Table 4/5?

**Clarity:**

The authors made a great deal of effort to explain the construction of the benchmark, and described each component of the benchmark in great detail, which is helpful.

**Correctness:**

- It would be helpful if the authors could include more description about the construction of the dataset, e.g., how is the graph set chosen?
- There is not much description of the model selection baselines presented in table 4, e.g., this includes MetaGL, which is from a prior work by the authors.

**Documentation:**

The GitHub code repository and the dataset folder are accessible, and readmes are provided to explain how to use these resources.

**Limitations:**

The authors discussed potential limitations and how future work could expand on this paper in Section 6.2. Potential negative societal impact of their work would be limited.

**Opportunities For Improvement:**

- It would be nice if the runtimes were reported for each model selection method in Table 6.
- I also wonder if the benchmark could be adopted for graph-level prediction tasks.

**Relation To Prior Work:**

The authors discussed the relation to prior work by the same set of authors in the introduction.

**Summary And Contributions:**

This paper constructs a large-scale evaluation benchmark to select graph-learning methods, including the underlying graph-learning algorithm and its hyperparameter settings.

This model selection problem is usually computationally expensive, e.g., requiring brute-force search or, more efficiently, with random search, grid search, Bayesian optimization, etc. These methods work in the offline setting, where one selects a model with some computational efforts before committing to which model to employ. This paper aims for the more ambitious goal of instantaneous model selection, meaning that the goal is to select the best model without querying model performances on the new datasets.

To achieve such instantaneous selection, the major components involve using historical model performances for the graph learning task of interest, and meta-graph features, quantifying similarities between graphs.

The contribution of this paper is introducing the GLEMOS benchmark--this includes many node/link classification evaluation tasks with many kinds of models, meta-graph features, model selection methods, and testbed.

Notably, the graph learning methods cover state-of-the-art graph neural networks and graph embedding methods and provide numerous options to adjust the hyperparameter settings for each algorithm.

The testbed includes sparse testbed, out-of-domain testbed, small-to-large testbed, and cross-task testbed. Experiments are provided to demonstrate various evaluations on each testbed.

---
The authors provided a detailed response to address the questions. Thus, I am raising my rating to Accept.

---

> ### Author Response · Authors · 2023-08-23
> **Response to Reviewer GJxQ**
>
> Thank you for your thoughtful comments. Please find our response to your comments below. We have also updated and highlighted the paper where necessary in light of your comment, which is marked in blue color in the revised paper.
>
> > It would be nice if the runtimes were reported for each model selection method in Table 6.
>
> We added additional results on the time cost of model selection methods in Appendix A.3.
> - First, in Figure 1 in Appendix, we report the runtime taken for predicting the best model given new graphs, which includes the time for generating meta-graph features, plus the time for a model selection method to infer the best model based on those features. For a majority of graphs, these two steps take at most five to six seconds in total.
> - Second, in Table 7 in Appendix, we report the training time of model selection algorithms with optimizable components. Results show that most of these learnable methods take less than a couple of minutes for training.
>
> > I also wonder if the benchmark could be adopted for graph-level prediction tasks.
>
> Yes, the benchmark could be adopted for graph-level prediction tasks. Datasets for graph-level tasks, such as graph classification, normally consist of multiple relatively small graphs, such as protein networks, where labels or target values are provided per graph. One way to incorporate graph-level tasks into the GLEMOS benchmark is to convert the set of graphs for a certain graph-level task into one graph by merging the individual graphs. That is, the resulting graph would consist of multiple disconnected components, where each component would correspond to one individual graph in the dataset. Then the meta-graph features extracted from the merged graph will be able to capture the distribution of characteristics of graphs in the graph-level dataset, and be used to model the similarity among graph datasets, in a way similar to how they are used for the node classification and link prediction tasks.
>
> > It would be helpful if the authors could include more description about the construction of the dataset, e.g., how is the graph set chosen?
>
> Thank you for the suggestion. Based on the suggestion, we have dedicated a section (Section 3.1) to discuss our principles for selecting the graphs and models included in the proposed benchmark.
>
> > There is not much description of the model selection baselines presented in table 4, e.g., this includes MetaGL, which is from a prior work by the authors.
>
> In Appendix B, we provide details of the model selection algorithms included in the GLEMOS benchmark.

---

> ### Author Response · Authors · 2023-08-23
> **Response to Reviewer GJxQ (Cont.)**
>
> > I wonder if the authors could discuss a little bit about some application scenarios of the benchmark; e.g., how would one imagine using it in a real-world application scenario?
>
> We expect two application scenarios of the proposed benchmark.
>
> **1. Practitioners of GL models.**
> Given a new graph dataset, practitioners aim to find the best GL model, which will perform a particular GL task (e.g., node classification) effectively on the given graph. This goal can be achieved by using the meta-graph features, model selection algorithms, and the performance records provided by the GLEMOS.
> More specifically, given a new graph, users will first generate meta-graph features using GLEMOS. Then the meta-features for the new graph will be provided to the model selection algorithm in GLEMOS, which have been pretrained using historical performance collections, and then the model selection algorithm will predict the best model for the new graph.
>
> **2. Researchers developing algorithms for GL model selection.**
> In this scenario, the goal is to develop new algorithms for the instantaneous GL model selection problem. The inference step (i.e., predicting the best GL model) in this scenario by the new algorithm would be the same as described above. In this case, GLEMOS facilitates algorithm development by providing benchmark testbeds, representative model selection algorithms, meta-graph features, as well as a diverse collection of performance records.
> Figure 2 shows steps involved with using GLEMOS for algorithm development and evaluation.
> - Estimating the effectiveness of a new algorithm is one of the most important goals of algorithm development. The new algorithm’s effectiveness can be evaluated using the multiple benchmark testbeds, which are designed to assess the performance of model selection algorithms in different usage scenarios.
> - Once a testbed is chosen, a model selection algorithm will be trained using the training data of the testbed, and evaluated using the test data of the testbed.
> - Since representative model selection algorithms are provided by GLEMOS, comparisons with existing methods in the same testbed can be done easily.
> - Also, GLEMOS provides a diverse collection of performance records over various graphs and GL models, and pre-generated meta-graph features, which saves a significant amount of time and resources needed to obtain these data for algorithm development.
>
> > How much computational cost is required to run the experiments reported in Table 4/5?
>
> For Table 4/5, we evaluated algorithms using 5-fold cross validation, and reported the averaged performance. In Table 7 in Appendix, we report the time taken for training model selection algorithms on the Fully-Observed testbed for the link prediction task, which is the setting that involves the largest amount of training data among all testbeds and GL tasks. The total training time for each algorithm would be five times the amount of time reported in Table 7 in Appendix. For the Sparse testbed, we evaluate using five different sparsity levels. Thus the training time for the Sparse testbed would be about five times that of the Fully-Observed testbed.

---

> > ### Comment · Reviewer_GJxQ · 2023-08-24
> > **Thanks**
> >
> > I have read the response and updated my preliminary review. Thanks for the detailed information.

---

> > > ### Author Response · Authors · 2023-08-25
> > >
> > > Thank you for the quick update, and thank you again for your review! It greatly contributed to improving the paper.

---

### Decision · Program_Chairs · 2023-09-22

**Decision:**

Accept (Poster)

**Comment:**

The paper introduces GLEMOS, a benchmark for instantaneous graph learning model selection. This benchmark evaluates 360 models on 460 graphs, covering both link prediction and node classification tasks. It offers a standardized and automated framework for selecting effective graph learning (GL) models without manual intervention. The proposed benchmark encompasses various evaluation settings, including fully observed, sparse, and out-of-domain scenarios, providing valuable performance data for over 200,000 model-dataset pairs.

GLEMOS incorporates an extensive collection of model performances and representative model selection techniques. The authors have designed over one thousand meta-graph features to measure graph similarities effectively. The experimental results demonstrate that GLEMOS outperforms existing GL model selection methods in terms of accuracy and efficiency, making it a valuable resource for the graph learning community.

The paper also offers a well-structured review of existing GL model selection methods, highlighting their limitations and presenting empirical observations from the proposed benchmark. Moreover, GLEMOS is designed to be easily extended with new models, graphs, and tasks, ensuring reproducibility and facilitating further research in the field. The authors provide datasets, code, and detailed documentation, enhancing the benchmark's accessibility and usability. Overall, GLEMOS addresses an important problem in automated model selection for graphs and represents a significant contribution to the graph learning community.